# Mitigating Covariate Shift in Behavioral Cloning via Robust Stationary Distribution Correction

**Seokin Seo**[1], **Byung-Jun Lee**[2,3], **Jongmin Lee**[4], **HyeongJoo Hwang**[1],
**Hongseok Yang**[1], **Kee-Eung Kim**[1]

[1]KAIST, [2]Korea University, [3]Gauss Labs Inc., [4]UC Berkeley

siseo@ai.kaist.ac.kr, byungjunlee@korea.ac.kr,
jongmin.lee@berkeley.edu, hjhwang@ai.kaist.ac.kr,
hongseok.yang@kaist.ac.kr, kekim@kaist.ac.kr

## Abstract

We consider offline imitation learning (IL), which aims to train an agent to imitate from the dataset of expert demonstrations without online interaction with the environment. Behavioral Cloning (BC) has been a simple yet effective approach to offline IL, but it is also well-known to be vulnerable to the covariate shift resulting from the mismatch between the state distributions induced by the learned policy and the expert policy. Moreover, as often occurs in practice, when expert datasets are collected from an arbitrary state distribution instead of a stationary one, these shifts become more pronounced, potentially leading to substantial failures in existing IL methods. Specifically, we focus on covariate shift resulting from arbitrary state data distributions, such as biased data collection or incomplete trajectories, rather than shifts induced by changes in dynamics or noisy expert actions. In this paper, to mitigate the effect of the covariate shifts in BC, we propose DrilDICE, which utilizes a distributionally robust BC objective by employing a stationary distribution correction ratio estimation (DICE) to derive a feasible solution. We evaluate the effectiveness of our method through an extensive set of experiments covering diverse covariate shift scenarios. The results demonstrate the efficacy of the proposed approach in improving the robustness against the shifts, outperforming existing offline IL methods in such scenarios.

## 1   Introduction

Imitation learning (IL) aims to recover the expert behavior from the dataset of demonstrations. The standard IL setting assumes that the imitator is allowed to interact with the environment during training, as it provides valuable information regarding state transitions. On the other hand, offline IL requires training without online interactions, reflecting scenarios where the interactions are either infeasible or expensive [5, 12, 13, 28]. Despite recent works on offline IL that explore scenarios involving supplementary datasets, such as suboptimal demonstrations [5, 14], behavioral cloning (BC) remains a compelling option in practice since BC does not require additional datasets except expert demonstrations. However, the efficacy of BC can be compromised when the imitator policy's behavior deviates from the underlying data distribution. This phenomenon, known as covariate shift, presents a significant challenge, often resulting in performance degradation. Consequently, a substantial body of IL research is dedicated to addressing this covariate shift issue [5, 24].

A common approach for solving offline IL problems is to use the so-called distribution matching objective or its variant [11, 12, 14, 15, 31], which aims to align the stationary distribution of the imitator policy with that of the expert policy. However, this approach crucially assumes that expert demonstrations are sampled from the stationary distribution of the expert policy. In practice, this

38th Conference on Neural Information Processing Systems (NeurIPS 2024).

assumption may break, that is, sampled expert demonstrations may be from a shifted version of the stationary distribution. For instance, expert demonstrations may be collected by first sampling states from a distribution that is different from the stationary distribution of the expert policy, and then labeling sampled states with expert actions. In this case, expert demonstrations are not samples from the stationary distribution. As a result, the offline IL algorithms based on the distribution-matching approach might perform badly in this covariate shift case.

In this paper, we address the issue of the covariate shift in the dataset of expert demonstrations, particularly when the data distribution does not match the stationary distribution of the expert policy. We begin by arguing that BC objective is more natural to consider the shift than the distribution matching objective. Then, inspired by the principle of distributionally robust optimization [2, 6, 20, 25], we propose an adversarial objective for offline IL that addresses covariate shifts in BC training. Instead of simply considering the worst-case with respect to all possible distributions on state-action pairs, our objective considers only those distributions that arise as the stationary distributions of policies, i.e., distributions that satisfy the Bellman flow constraint. Leveraging the techniques from the stationary distribution correction ratio (DICE) algorithm, we introduce DrilDICE, which efficiently solves our optimization problem and computes an imitator robust to covariate shifts of the dataset. In addition, we suggest the practical covariate shift scenarios that may arise in offline IL applications. Under those problem settings, we compare our approach with baselines and demonstrate that our approach can imitate the agent robust to covariate shift of our interests.

## 2 Background and Related Work

### 2.1 Offline Imitation Learning with Arbitrary State Distributions

We consider a Markov decision process (MDP) without rewards, which is defined as a tuple of $\langle \mathcal{S}, \mathcal{A}, T, p_0, \gamma \rangle$ where $\mathcal{S}$ is a state space, $\mathcal{A}$ is an action space, $T : \mathcal{S} \times \mathcal{A} \to \Delta(\mathcal{S})$ is a transition distribution, $p_0 \in \Delta(\mathcal{S})$ is an initial state distribution and $\gamma \in [0, 1)$ is a discounted factor. We focus on the class of deterministic policies $\Pi = \{\pi : \mathcal{S} \to \mathcal{A}\}$. Given a policy $\pi \in \Pi$, the stationary distribution $d_\pi$ of $\pi$ is defined as $d_\pi(s, a) = (1 - \gamma) \sum_{t=0}^{\infty} \gamma^t \Pr(s_t = s, a_t = a)$ where $s_0 \sim p_0, a_t = \pi(s_t), s_{t+1} \sim \mathcal{T}(s_t, a_t)$. For convenience, we use $\pi_E$ to denote the policy of an expert and write simply $d_E$ for the stationary distribution $d_{\pi_E}$ of $\pi_E$.

Assuming the existence of an expert policy $\pi_E \in \Pi$, the goal of imitation learning (IL) is to recover $\pi_E$ by utilizing some demonstrations of the expert. Specifically, offline IL prohibits online interactions and relies solely on a given offline dataset that consists of demonstrations of the expert. We consider that the demonstration dataset $\mathcal{D}$ is the collection of $(s, a, s')$ triplets where $s$ is sampled from an arbitrary state distribution $d_D(s)$, $a$ is determined by $\pi_E(s)$ and $s'$ is sampled from $T(s'|s, a)$. Note that conventional IL approaches assume that $d_D$ is close to the expert's state distribution $d_E$. However, unlike previous studies, we focus the imitation learning with an arbitrary state distribution $d_D$, without assuming that $d_D$ to be a state stationary distribution of any policy.

As we mentioned in the introduction, one common approach for offline IL is to use the distribution matching objective or its variant [11, 12, 14, 15] and to find an imitator policy whose stationary distribution matches that of the expert policy. But as we will show later in the paper, the algorithms based on this approach do not perform well when there is a covariate shift in the given demonstration dataset $d_D(s, a)$ in a way explained in the previous paragraph. In the paper, we propose an offline IL algorithm that achieves good performance in the presence of such a covariate shift by using a version of distributionally robust optimization.

### 2.2 Covariate Shift in Imitation Learning

Covariate shift is a widely used term in machine learning [1, 7]. Traditionally, it refers to the phenomenon where the distribution of covariates (inputs) during training differs from that during testing, i.e. given a covariate variable $X \in \mathcal{X}$ and a response variable $Y \in \mathcal{Y}$,

$$p_{\text{train}}(Y|X = x) = p_{\text{test}}(Y|X = x) \quad \forall x \in \mathcal{X}, \qquad p_{\text{train}}(X) \neq p_{\text{test}}(X).$$

The distribution shift that is considered in this paper has this form where state $s$ and action $a$ correspond to $X$ and $Y$, and $p_{\text{train}}$ and $p_{\text{test}}$ correspond to the distribution $d_D$ of the given demonstration dataset and the stationary distribution $d_E$ of the expert policy $\pi_E$.

One significant branch of research addressing the issue of covariate shift is distributionally robust optimization (DRO). DRO involves considering an uncertainty set of the distribution and optimizing for the worst-case loss within this set [6, 20]. Recent studies have proposed method to attain the worst-case supremum by using kernel methods [25] or the integral probability metric such as Wasserstein metric [10]. Additionally, in the field of imitation learning, DRO has been applied to achieve robust learning in the presence of noisy experts [2] or transition shifts [19].

A closely related problem studied in IL is robust imitation learning, which focuses on learning a policy that can tolerate various shifts in the transition distribution $T$ of the MDP [4, 19]. Another related scenario is the noisy expert scenario, where the demonstration dataset is corrupted by noise, resulting in the distribution $d_D$ of the dataset being a noisy version of $d_E$ [6, 23]. These issues are usually addressed by using robust optimization techniques [6, 19], which are also utilized in our approach. However, it is important to note that the types of distribution shifts considered in our work and in prior researches differ within the context of offline IL.

To emphasize, we are not focused on addressing covariate shifts induced by transition shifts or noisy demonstrations. Instead, our work investigates a type of covariate shift where the dataset still consists of expert actions, but states are not sampled from the stationary distribution of the expert policy: the distribution $d_D(s)$ of the samples in the given demonstration dataset differs from $d_E(s)$, but both distributions share the expert policy given states, i.e., $\pi_D(s) = \pi_E(s)$ for all $s \in \mathcal{S}$.

## 3 Mitigating Covariate Shift in BC

### 3.1 Robustness to Covariate Shift

Assume that we have a free access to $\pi_E$, such that we can optimize our trained policy $\pi$ to be close to it on any state. We can write the BC objective for optimizing $\pi$ as:

$$\min_\pi \ \mathbb{E}_{s \sim d_E}[\ell(\pi(s), \pi_E(s))] \tag{1}$$

for some supervised learning loss $\ell$ (e.g. squared error, cross-entropy, ...). Previous studies on imitation learning have assumed that $d_D \approx d_E$. Under this assumption, BC approach of:

$$\min_\pi \mathbb{E}_{s \sim d_D}[\ell(\pi(s), \pi_E(s))] \tag{2}$$

does show strong performance if we have sufficient data. Otherwise, if the expert data is not sufficient, we can also incorporate transition information and benefit from the distribution matching (DM) approach, which aims to get a stationary distribution $d_{\mathrm{DM}}$ that is close to data distribution $d_D$ with the following objective:

$$d_{\mathrm{DM}} := \underset{d \in \Delta(\mathcal{S})}{\arg\max} \ - \mathbb{D}(d(s,a) \| d_D(s,a))$$

$$\text{s. t.} \ \sum_a d(s,a) = (1-\gamma)\rho_0(s) + \gamma \sum_{\bar{s},\bar{a}} T(s|\bar{s},\bar{a})d(\bar{s},\bar{a}) \quad \forall s \in \mathcal{S}$$

where $\mathbb{D}$ is any divergence between two probability distributions. The solution of the optimization problem $d_{\mathrm{DM}}$ can be used for the state distribution for BC instead of $d_E$, and it is widely known that this can complement lack of data with additional transition information, leading to improved performance. Nevertheless, under harsh covariate shift, i.e., when the state distribution of $d_D$ diverges from that of $d_E$, both two approaches above can fail miserably, as there is no guarantee on improving policy performance when $d_E$ and $d_D$ are significantly different. See Section A for an example of failure cases.

To overcome the covariate shift, we adopt a distributionally robust objective, i.e., for an arbitrary distribution $d$, we know $\mathbb{E}_{s \sim d_E}[\ell(\pi(s), \pi_E(s))] \leq \max_{d \in \Delta(S)} \mathbb{E}_{s \sim d}[\ell(\pi_E(s), \pi(s))]$, which holds with $d_\pi$ as well, and we can instead optimize for

$$\min_\pi \max_{d \in \mathcal{Q}} \mathbb{E}_{s \sim d}[\ell(\pi(s), \pi_E(s))], \tag{3}$$

which corresponds to an upper-bound minimization for optimizing $\pi$ with respect to some uncertainty set $\mathcal{Q} \subseteq \Delta(\mathcal{S})$. However, when $\mathcal{Q}$ is choosed as $\Delta(\mathcal{S})$, this objective can easily lead to overly pessimistic solutions, as we seek for any extreme state distribution $d$ that maximizes the policy loss, and it will easily place all its probability on a single state. Hence, a choice of $\mathcal{Q}$ is crucial to obtain an appropriate solution of distributionally robust optimization.

## 3.2 Towards Less Pessimistic Robust Objective

For notational convenience, denote $C_\pi(s) := \ell(\pi(s), \pi_E(s))$. In the objective (3), the worst-case policy loss is considered among an uncertainty set $\mathcal{Q}$. However, when $\mathcal{Q} = \Delta(\mathcal{S})$, it would be too pessimistic and the objective can be improved when we consider a smaller choice of $\mathcal{Q}$. To keep the fact that it is an upper-bound minimization for Eq. 1, which is the loss with $d_E$, the expert state distribution $d_E$ should still be contained in $\mathcal{Q}$, even if we aim to use a smaller set.

To facilitate this, we impose two constraints on $\mathcal{Q}$ with respect to a state distribution $d$: (1) $d$ should satisfy Bellman flow constraint, (2) $d$ should be close enough to $d_D$ to prevent $d$ diverging too far from $d_D$[1]. The first constraint is a constraint that should be satisfied by $d_E$, and thus it effectively reduces the potential distribution set for $d$ if we impose it properly. The second constraint may not seem necessary, but as we estimate the policy loss from finite samples from $d_D$, diverging too far from $d_D$ will reduce effective number of samples. Then, by introducing $f$-divergence regularization to enforce the second constraint, we consider the following constrained optimization problem:

$$
\begin{aligned}
\min_\pi \quad & \max_{d \in \Delta(\mathcal{S})} \quad \mathbb{E}_{s \sim d}[C_\pi(s)] - \alpha \mathbb{D}_f(d(s,a) \| d_D(s,a)) \\
& \text{s.t.} \quad \sum_a d(s,a) = (1-\gamma)\rho_0(s) + \gamma \sum_{\bar{s}, \bar{a}} T(s|\bar{s}, \bar{a}) d(\bar{s}, \bar{a}) \quad \forall s \in \mathcal{S}
\end{aligned} \tag{4}
$$

where $\mathbb{D}_f$ denotes $f$-divergence with a convex function $f$. We develop a practical algorithm in a stationary distribution correction ratio estimation (DICE) framework style. From Eq. 4, we can take Lagrangian $\nu$ to solve the inner constrained optimization, i.e.

$$
\begin{aligned}
& \max_{d \in \Delta(\mathcal{S})} \min_\nu \sum_{s,a} d(s,a) C_\pi(s) - \alpha \mathbb{D}_f(d(s,a) \| d_D(s,a)) \\
& \quad + \sum_s \nu(s)\left[(1-\gamma)\rho_0(s) + \gamma \sum_{\bar{s}, \bar{a}} T(s|\bar{s}, \bar{a}) d(\bar{s}, \bar{a}) - \sum_a d(s,a)\right] \\
& = \max_{w \in \mathcal{W}} \min_\nu (1-\gamma) \mathbb{E}_{s \sim \rho_0}[\nu(s)] + \mathbb{E}_{(s,a,s') \sim d_D}\left[-\alpha f\left(w(s,a)\right) + w(s,a) e_{\pi,\nu}(s,a,s')\right]
\end{aligned} \tag{5}
$$

where $e_{\pi,\nu} := \left(C_\pi(s) + \gamma \sum_{s'} T(s'|s,a)\nu(s') - \nu(s)\right)$, $w(s,a) := \frac{d(s,a)}{d_D(s,a)}$ and $\mathcal{W}$ is a class of functions $w : \mathcal{S} \times \mathcal{A} \to \mathbb{R}_{\geq 0}$.

By Slater's condition [3], since the strong duality holds on Eq. 5, we can change the minimax into a maximin optimization as follows:

$$
\min_\nu \max_{w \in \mathcal{W}} (1-\gamma) \mathbb{E}_{s \sim \rho_0}[\nu(s)] + \mathbb{E}_{(s,a,s') \sim d_D}\left[-\alpha f\left(w(s,a)\right) + w(s,a) e_{\pi,\nu}(s,a,s')\right], \tag{6}
$$

and the closed-form solution $w^*$ can be obtained by solving for the inner maximization.

**Proposition 1.** *(Lee et al. [16]) The closed-form solution for the inner maximization of Eq. (6) is*

$$
w^*_{\pi,\nu}(s,a) = \max\left(0, (f')^{-1}\left(\frac{e_{\pi,\nu}(s,a)}{\alpha}\right)\right) \quad \forall s, a \tag{7}
$$

After obtaining $w^*_{\pi,\nu}$, the minimax optimization (6) is converted into a minimization problem with respect to $\nu$. As a result, each objective for $\nu, \pi$ can be summarized as follows:

$$
\begin{aligned}
\min_\nu \quad & (1-\gamma)\mathbb{E}_{s \sim \rho_0}[\nu(s)] + \mathbb{E}_{(s,a,s') \sim d_D}\left[-\alpha f\left(w^*_{\pi,\nu}(s,a)\right) + w^*_{\pi,\nu}(s,a) e_{\pi,\nu}(s,a,s')\right] \\
\min_\pi \quad & \mathbb{E}_{(s,a) \sim d_D}\left[w^*_{\pi,\nu}(s,a) C_\pi(s)\right]
\end{aligned} \tag{8}
$$

Hence, the problem can be practically addressed by alternatively optimizing $\nu$ and $\pi$ following recent developments in DICE approaches [16, 17]. We call this distributionally robust optimization approach to covariate shift in offline IL as DrilDICE (**D**istributionally **R**obust **I**mitation **L**earning via **DICE**).

---

[1]Note that this constraint is analogous to the use of a robustness radius used in [19].

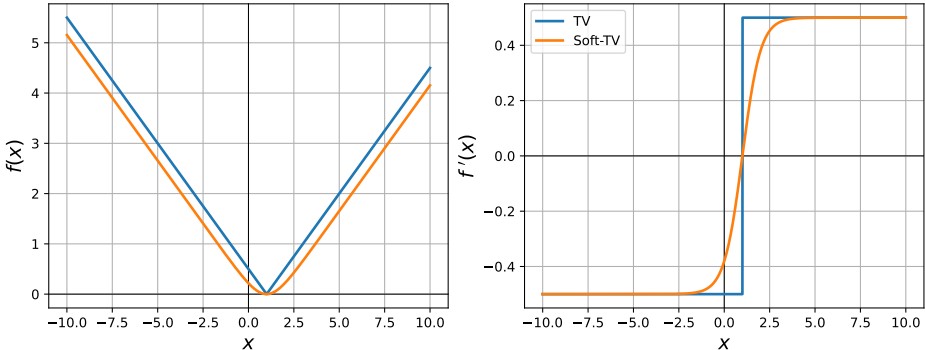

Figure 1: Illustration of soft TV-distance. (left) $f$ functions, (right) corresponding derivatives $f'$.

### 3.3 Soft TV-Distance

To facilliate a fair comparison with the uncertainty set defined by total variation (TV) distance, as utilized in [19], we introduce the soft-TV distance as a specific choice of $f$-divergence of DrilDICE. It is important to note that **Proposition 1** requires $(f')^{-1}$, the invertable derivative of $f$. However, the generator function of TV-distance $f_{\text{TV}}(x) := \frac{1}{2}|x - 1|$, lacks an invertible derivative as illustrated in Figure 1, rendering the direct application of TV distance in DrilDICE challenging.

We technically overcome this limitation by relaxing the derivative function of TV-distance. Given that the derivative function of TV-distance manifests as a step function, we choose to relax this function by employing the $\tanh$ function. Specifically, we utilize the log-cosh function [22] to $f$ as follows:

$$f_{\text{SoftTV}}(x) = \frac{1}{2}\log(\cosh{(x-1)}), \quad (f'_{\text{SoftTV}})^{-1}(y) = \tanh^{-1}{(2y)} + 1. \tag{9}$$

By plugging $f'_{\text{SoftTV}}$ into Eq. 7, we can obtain the closed-form solution $w^*_{\pi,\nu}$ of the inner maximization objective (6) tailored to a specific choice of $f_{\text{Soft-TV}}$ as follows:

$$w^*_{\pi,\nu}(s, a) = \text{ReLU}\left(\tanh^{-1}\left(\frac{2e_{\pi_\nu}(s, a, s')}{\alpha}\right) + 1\right) \tag{10}$$

where $\text{ReLU}(x) := \max(x, 0)$. This choice of $f$ enables DrilDICE to obtain a closed form solution of the inner maximization problem while maintaining similar properties of TV-distance. For other possible choice of $f$-divergence in DrilDICE, see Section B in the supplementary material.

## 4 Experiments

### 4.1 Comparison on Baselines Objectives

To provide clear descriptions about the baselines used for comparison in our experiments, we summarize the relevant objectives in Table 1.

**Adversarial Weighted BC (AW-BC)**  We refer to adversarial weighting by following the terminology from [27] as the minimax objective without constraints. As we discussed in Section 3.2, the adversarial weighting objective is also an upper-bound of the target objective. However, it tends to be overly pessimistic because it considers the entire state distribution space, including distributions that do not correspond to the stationary distribution of any policy.

**Distributionally Robust BC (DR-BC)**  To adjust the level of the robustness, DR-BC [19] can be employed in this context. Despite originally designed to address transition shifts, DR-BC remains relevant to our scenario since it considers the uncertainty set over state distributions by adopting a robustness radius hyperparameter $\rho$. Notably, the $f$-divergence constraint of DrilDICE is functionally analogous to the robustness radius constraint $D_{\text{TV}}(d\|d_D) \leq \rho$. However, our approach, which incorporates Bellman flow constraints, addresses a more restricted set when constrained by the equivalent radius level, thereby offering a tighter upper-bound to Eq. 1.

| Functionality | Objective |
|---|---|
| Distribution matching | $\min_{\pi} \mathbb{E}_{s \sim d_{\text{DM}}}[C_\pi(s)] \quad \text{s.t.} \quad d_{\text{DM}} = \arg\max_{d \in \mathcal{Q}} -\mathbb{D}_f(d(s,a) \| d_D(s,a))$ |
| AW-BC objective | $\min_{\pi} \max_{d \in \Delta S} \; \mathbb{E}_{s \sim d}[C_\pi(s)]$ |
| DR-BC objective | $\min_{\pi} \max_{d : D_{\text{TV}}(d \| d_D) \leq \rho} \; \mathbb{E}_{s \sim d}[C_\pi(s)]$ |
| Best-case weighting | $\min_{\pi} \max_{d \in \bar{\mathcal{Q}}} \; \mathbb{E}_{s \sim d}[-C_\pi(s)] - \alpha \mathbb{D}_f(d(s,a) \| d_D(s,a))$ |
| Worst-case weighting (Ours) | $\min_{\pi} \max_{d \in \bar{\mathcal{Q}}} \; \mathbb{E}_{s \sim d}[C_\pi(s)] - \alpha \mathbb{D}_f(d(s,a) \| d_D(s,a))$ |

Table 1: Objective comparisons for related approaches. Denote a stationary distributions class as $\bar{\mathcal{Q}}$, i.e., $\bar{\mathcal{Q}} := \{d \in \Delta(\mathcal{S}) : d(s) = (1-\gamma)\rho_0(s) + \gamma \sum_{\bar{s}, \bar{a}} T(s|\bar{s}, \bar{a}) d(\bar{s}, \bar{a}) \forall s \in \mathcal{S}\}$

**Best-case Weighting** If the sign of the cost function in the worst-case weighting objective is converted, the objective seeks to find a cost-minimizing stationary distribution that is close to the data collection policy. We call this objective the best-case weighting. When $d_E \in \mathcal{Q}$, the best-case weighting minimizes the lower bound of the target objective since $\min_{d \in \mathcal{Q}} \mathbb{E}_d[C_\pi(s)] \leq \mathbb{E}_{d_E}[C_\pi(s)]$, which is not relevant to minimize the target objective. Despite there is no direct connection, to ensure that performance of our method is not due to side effects of algorithm's implementation, we include this method for performance comparison in the following experiment section.

## 4.2 Toy Domain Experiment: Four Rooms domain with Imbalanced Datasets

### 4.2.1 Experiment Settings

**Four Rooms environment** Four Rooms is a grid-world environment which aims to find a path from a starting state to a goal state. Each cell in the grid represents a state and the starting state and the goal state are marked by the orange and the green box in Figure 2 respectively. The agent can choose one of 4 actions from each state: UP, DOWN, LEFT, RIGHT, and the rooms are numbered as shown in Figure 2. We pre-collect expert dataset $D_E$ by using a deterministic expert policy $\pi_E$ and online interactions.

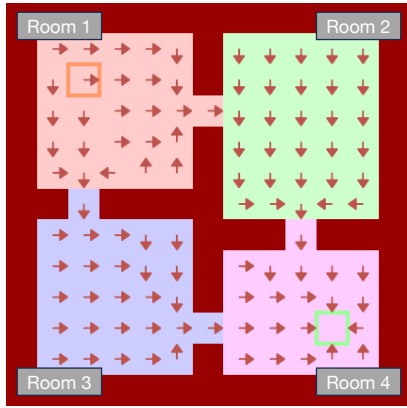

Figure 2: Four Rooms environment and a deterministic expert (red arrows).

**Covariate shift scenarios** To investigate that our approach can effectively addresses the covariate shift, we need the dataset that notably deviates from the expert stationary distribution $d_E$. A possible realistic scenario of the dataset deviation occurs when data collectors gather data at different frequencies for each state or action. For example, if a data collection device (e.g. cameras or sensors) operates at different recording frequencies in each room, the frequency of observing a room in the dataset will not match $d_E$. To simulate this scenario, we design datasets where the marginal room (or action) distribution of dataset deviates from $d_E$. By manipulating the marginal proportion of a predetermined variable in dataset, we generate a deviated dataset from the original dataset. We consider a total of 8 problem settings by manipulating the marginal distribution of four rooms and four actions. In each scenario, we set a marginal distribution $p_i(u)$ of the variable to be manipulated $u$ to a certain fixed probability and resample transitions to configure the dataset $D_i$ through the following process. (See Section C in the supplementary material)

$$D_i = \{(s, a) | u \sim p_i(u), s \sim D_E(s|u), a = \pi_E(s)\}$$

**Implementations[2] and baselines** Since we are not interested in trivial scenarios where the policy can easily minimize the supervised loss over entire state space to zero (i.e. $\max_{s \in \mathcal{S}} C_\pi(s) = 0$), we conduct experiments using function approximators instead of a tabular setting. We initialize $\pi_0$ for the cost function $C_\pi$ as BC policy. To solve optimization problems, each algorithm is implemented

---

[2] https://github.com/tzs930/drildice

| | Scenario | Metrics | BC | DemoDICE | AW-BC | DR-BC | OptiDICE-BC | DrilDICE (Ours) |
|---|---|---|---|---|---|---|---|---|
| Room Marginal Manipulation | Room 1 | Normalized score | 90.84 ± 0.69 | 91.62 ± 0.65 | 90.84 ± 0.58 | 91.38 ± 0.73 | 94.30 ± 0.41 | **95.04 ± 0.48** |
| | | Worst-25% | 63.36 ± 2.74 | 66.48 ± 2.59 | 63.36 ± 2.31 | 65.68 ± 2.84 | 77.20 ± 1.64 | **80.16 ± 1.92** |
| | | Target 0-1 loss ($\times 10^2$) | 8.69 ± 0.06 | 8.28 ± 0.06 | 7.92 ± 0.05 | 8.42 ± 0.55 | **7.85 ± 0.05** | 8.31 ± 0.07 |
| | Room 2 | Normalized score | 89.16 ± 1.07 | 89.72 ± 1.00 | 89.20 ± 0.78 | 89.28 ± 1.08 | 94.06 ± 0.65 | **94.44 ± 0.62** |
| | | Worst-25% | 57.84 ± 3.90 | 59.36 ± 3.83 | 57.28 ± 2.88 | 58.32 ± 3.92 | 76.24 ± 2.61 | **77.76 ± 2.47** |
| | | Target 0-1 loss ($\times 10^2$) | 10.75 ± 0.15 | 10.51 ± 0.14 | 9.34 ± 0.09 | 10.72 ± 1.03 | 6.85 ± 0.08 | **6.46 ± 0.07** |
| | Room 3 | Normalized score | 88.50 ± 1.27 | 89.66 ± 1.21 | 88.68 ± 1.16 | 88.70 ± 1.26 | 94.20 ± 0.98 | **95.04 ± 0.86** |
| | | Worst-25% | 55.28 ± 4.74 | 59.60 ± 4.56 | 55.52 ± 4.40 | 55.92 ± 4.75 | 76.80 ± 3.92 | **80.16 ± 3.44** |
| | | Target 0-1 loss ($\times 10^2$) | 13.85 ± 0.16 | 13.22 ± 0.15 | 11.75 ± 0.16 | 13.74 ± 1.11 | 9.64 ± 0.14 | **8.51 ± 0.13** |
| | Room 4 | Normalized score | 90.94 ± 0.75 | 91.34 ± 0.70 | 90.62 ± 0.75 | 90.94 ± 0.75 | 94.26 ± 0.54 | **94.92 ± 0.37** |
| | | Worst-25% | 64.00 ± 2.89 | 65.36 ± 2.81 | 62.72 ± 2.91 | 64.00 ± 2.89 | 77.04 ± 2.15 | **79.68 ± 1.49** |
| | | Target 0-1 loss ($\times 10^2$) | 9.30 ± 0.09 | 9.34 ± 0.08 | 8.70 ± 0.08 | 9.30 ± 6.20 | 8.21 ± 0.07 | **7.97 ± 0.05** |
| Action Marginal Manipulation | Action UP | Normalized score | 84.96 ± 1.33 | 87.20 ± 1.17 | 84.94 ± 1.27 | 84.96 ± 1.33 | 92.06 ± 0.69 | **93.22 ± 0.61** |
| | | Worst-25% | 43.92 ± 4.43 | 51.04 ± 4.04 | 43.44 ± 4.18 | 43.92 ± 4.43 | 68.32 ± 2.73 | **72.88 ± 2.43** |
| | | Target 0-1 loss ($\times 10^2$) | 13.80 ± 0.18 | 11.83 ± 0.16 | 12.64 ± 0.16 | 13.80 ± 1.26 | 8.29 ± 0.09 | **8.18 ± 0.08** |
| | Action DOWN | Normalized score | 89.96 ± 0.96 | 91.28 ± 0.84 | 90.52 ± 0.90 | 89.96 ± 0.96 | 93.62 ± 0.62 | **94.60 ± 0.39** |
| | | Worst-25% | 60.80 ± 3.45 | 65.36 ± 3.27 | 62.24 ± 3.53 | 60.80 ± 3.45 | 74.48 ± 2.47 | **78.40 ± 1.55** |
| | | Target 0-1 loss ($\times 10^2$) | 10.45 ± 0.13 | 9.53 ± 0.12 | 8.79 ± 0.12 | 10.45 ± 0.89 | 7.73 ± 0.09 | **7.13 ± 0.06** |
| | Action LEFT | Normalized score | 90.18 ± 1.11 | 90.32 ± 1.10 | 90.80 ± 1.11 | 90.18 ± 1.11 | 91.86 ± 1.03 | **92.62 ± 0.95** |
| | | Worst-25% | 61.60 ± 4.15 | 62.16 ± 4.10 | 64.40 ± 4.02 | 61.60 ± 4.15 | 68.32 ± 3.73 | **71.36 ± 3.38** |
| | | Target 0-1 loss ($\times 10^2$) | 11.40 ± 1.02 | 11.16 ± 0.15 | 10.56 ± 0.15 | 11.42 ± 1.02 | 9.94 ± 0.14 | **9.16 ± 0.12** |
| | Action RIGHT | Normalized score | 93.04 ± 0.63 | 93.12 ± 0.62 | 92.76 ± 0.51 | 93.44 ± 0.60 | 94.46 ± 0.44 | **94.52 ± 0.44** |
| | | Worst-25% | 72.16 ± 2.52 | 72.48 ± 2.48 | 71.04 ± 2.02 | 73.76 ± 2.38 | 77.84 ± 1.77 | **78.08 ± 1.77** |
| | | Target 0-1 loss ($\times 10^2$) | 7.07 ± 0.06 | 7.04 ± 0.06 | 6.58 ± 0.04 | 6.62 ± 4.04 | **6.48 ± 0.05** | 6.56 ± 0.05 |

Table 2: Comparison of different methods across manipulated datasets in Four Rooms environment. ($p(u) = 0.4$) Each experiment is repeated with 50 times and the average values with their standard errors are reported. The best average values are highlighted in bold.

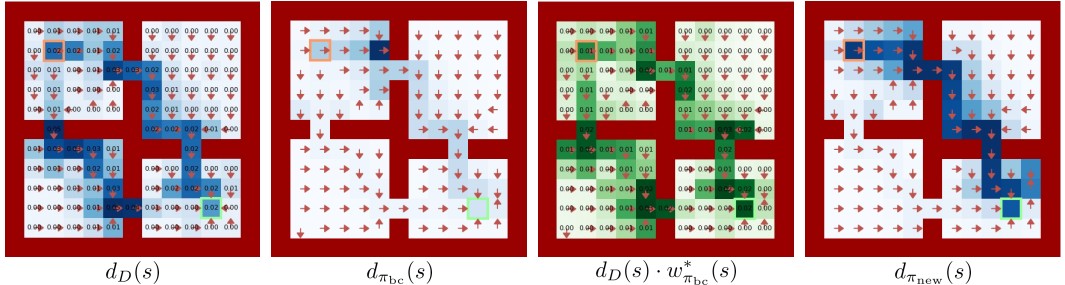

$d_D(s)$    $d_{\pi_{\text{bc}}}(s)$    $d_D(s) \cdot w^*_{\pi_{\text{bc}}}(s)$    $d_{\pi_{\text{new}}}(s)$

Figure 3: Visualizations of policy behaviors and weights. (a) $d_D(s)$ : dataset state distribution, (b) $d_{\pi_{\text{bc}}}(s)$ : behavior of BC policy, (c) $d(s)w^*_{\pi_{\text{bc}}}(s)$: corrected state distribution that maximizes $C_\pi$, (d) behavior of $d_{\pi_{\text{new}}}(s)$: DrilDICE policy (BC weighted with $w^*_{\pi_{\text{bc}}}(s)$).

with a two-step procedure: (1) optimize each objective by using the cost $C_{\pi_0}$ determined by an initial policy $\pi_0$ and obtain $w^*_{\pi_0}$, (2) train $\pi$ with the weighted BC. For policy and weight models, we use linear approximators with RBF features, which exploit distances from representative points. For more details, see Section C. For choices of $f$-divergence for OptiDICE-BC and DrilDICE, we select KL-divergence instead of the soft TV-distance for a simplicity of convex optimization solver implementation. We consider following baselines:

- **BC**: a standard behavioral cloning without any regularization.
- **DemoDICE** [14] : a representative distribution matching approach to offline IL. We compare a special case of DemoDICE that does not exploits supplementary datasets.
- **AW-BC**: adversarial weighting method without Bellman flow constraints and robustness radius.
- **DR-BC** [19]: distributionally robust BC method without Bellman flow constraints.
- **OptiDICE-BC** [16] : a representative method as the best-case weighting.

**Evaluation metrics**    The following metrics are measured with 100 episodes for each trained policy.

- **Normalized score**: a normalized episode return that scales from 0 (random score) up to 100 (expert score) averaged by 100 episodes.
- **Worst-25% performance** : the normalized scores averaged by the worst 25% episodes.
- **Target 0-1 loss**: the averaged 0-1 loss (i.e. $\mathbb{E}_{d_\pi}[\mathbb{I}(\pi(s) \neq \pi_E(s))]$)[3]

---

[3]For convenience, we evaluate the expected loss over $d_\pi$ instead of the original target $d_E$. It is important to note that DRO objectives are also upper bounds for this expected loss if $d_\pi \in \mathcal{Q}$.

| | Task | $p(D_1)$ | BC | DemoDICE | AW-BC | DR-BC | OptiDICE-BC | DrilDICE (Ours) |
|---|---|---|---|---|---|---|---|---|
| **Rebalanced by state** | hopper | 0.1 | $24.7 \pm 4.2$ | $25.8 \pm 3.2$ | $17.9 \pm 2.0$ | $27.0 \pm 4.3$ | $12.7 \pm 1.3$ | $\mathbf{52.2 \pm 5.6}$ |
| | | 0.5 | $35.4 \pm 4.1$ | $37.6 \pm 15.1$ | $24.6 \pm 1.7$ | $36.7 \pm 3.6$ | $8.6 \pm 2.0$ | $\mathbf{67.1 \pm 8.2}$ |
| | | 0.9 | $11.2 \pm 2.5$ | $15.0 \pm 3.4$ | $13.4 \pm 4.8$ | $27.4 \pm 4.9$ | $10.4 \pm 1.8$ | $\mathbf{36.4 \pm 6.1}$ |
| | walker2d | 0.1 | $18.9 \pm 4.1$ | $15.1 \pm 2.4$ | $7.3 \pm 1.2$ | $14.7 \pm 3.3$ | $4.9 \pm 1.1$ | $\mathbf{51.6 \pm 8.2}$ |
| | | 0.5 | $22.9 \pm 3.1$ | $30.2 \pm 3.7$ | $27.8 \pm 2.8$ | $45.1 \pm 10.0$ | $8.1 \pm 0.4$ | $\mathbf{73.7 \pm 5.4}$ |
| | | 0.9 | $30.4 \pm 7.1$ | $21.6 \pm 3.3$ | $35.8 \pm 5.0$ | $46.0 \pm 8.4$ | $7.7 \pm 0.5$ | $\mathbf{77.6 \pm 5.5}$ |
| | halfcheetah | 0.1 | $49.3 \pm 5.2$ | $38.1 \pm 4.6$ | $\mathbf{53.2 \pm 4.3}$ | $32.9 \pm 3.8$ | $7.1 \pm 1.5$ | $52.5 \pm 3.6$ |
| | | 0.5 | $38.0 \pm 3.1$ | $43.9 \pm 3.4$ | $32.6 \pm 1.9$ | $26.2 \pm 4.9$ | $6.1 \pm 1.2$ | $\mathbf{55.0 \pm 3.3}$ |
| | | 0.9 | $15.5 \pm 3.1$ | $5.0 \pm 1.1$ | $7.8 \pm 1.2$ | $9.0 \pm 3.3$ | $1.0 \pm 1.1$ | $\mathbf{22.3 \pm 2.9}$ |
| **Rebalanced by action** | hopper | 0.1 | $29.7 \pm 4.0$ | $26.8 \pm 3.0$ | $24.6 \pm 4.8$ | $25.9 \pm 2.5$ | $11.7 \pm 2.1$ | $\mathbf{56.6 \pm 11.9}$ |
| | | 0.5 | $26.4 \pm 4.9$ | $34.3 \pm 4.4$ | $30.8 \pm 2.3$ | $35.1 \pm 5.4$ | $11.8 \pm 1.1$ | $\mathbf{73.8 \pm 3.6}$ |
| | | 0.9 | $30.5 \pm 3.6$ | $16.5 \pm 1.4$ | $19.3 \pm 2.9$ | $36.6 \pm 2.3$ | $19.4 \pm 2.8$ | $\mathbf{49.0 \pm 12.3}$ |
| | walker2d | 0.1 | $23.6 \pm 5.1$ | $22.6 \pm 6.5$ | $20.4 \pm 4.2$ | $31.2 \pm 4.2$ | $7.3 \pm 0.5$ | $\mathbf{70.6 \pm 3.2}$ |
| | | 0.5 | $32.3 \pm 6.7$ | $33.8 \pm 7.5$ | $25.7 \pm 3.5$ | $30.5 \pm 3.9$ | $6.4 \pm 1.0$ | $\mathbf{72.1 \pm 8.7}$ |
| | | 0.9 | $16.9 \pm 2.8$ | $12.0 \pm 1.0$ | $16.9 \pm 2.5$ | $37.6 \pm 9.0$ | $4.6 \pm 1.0$ | $\mathbf{69.5 \pm 8.5}$ |
| | halfcheetah | 0.1 | $41.9 \pm 4.8$ | $34.9 \pm 3.1$ | $41.7 \pm 3.9$ | $27.5 \pm 1.0$ | $8.4 \pm 3.4$ | $\mathbf{56.4 \pm 4.6}$ |
| | | 0.5 | $45.8 \pm 4.5$ | $32.5 \pm 2.0$ | $30.7 \pm 2.3$ | $33.4 \pm 6.5$ | $4.6 \pm 0.8$ | $\mathbf{60.8 \pm 1.6}$ |
| | | 0.9 | $25.9 \pm 3.4$ | $8.8 \pm 3.1$ | $14.2 \pm 0.7$ | $12.1 \pm 2.0$ | $0.6 \pm 0.7$ | $\mathbf{29.2 \pm 4.6}$ |

Table 3: Performance comparison on Scenario 1 (rebalanced dataset). $p(D_1)$ determines the proportion of dataset $D_1$, which is close to the representative point. Each experiment is repeated with 5 times and the average normalized scores with their standard errors are reported. The highest mean performance scores are highlighted in bold.

## 4.3 Results

As depicted Table 2, DrilDICE outperformed baselines in the Four Rooms environment under various covariate shift scenarios. Our approach achieved the highest normalized scores across all scenarios and consistently demonstrated superior performance in the worst-25% performance. In scenarios involving Room 3, where the original dataset $D_E$ had only 8.9% coverage, the probability of observing states in Room 3 was increased to 40%, causing significant covariate shifts. This led to a significant degradation in the robust performance of BC (worst-25%). However, DrilDICE successfully improved the worst-25% performance in Room 3 to levels comparable to other scenarios, demonstrating its robustness to covariate shifts induced by dataset deviations.

Figure 3 illustrates the behaviors of BC and DrilDICE in the Room 3 scenario. With an increased dataset proportion visiting Room 3, as shown in Figure 3-(a), $\pi_{bc}$ performs accurately in Room 3, predicting the correct expert actions in nearly all states (26 out of 27) in Room 3. However, its performance remains suboptimal in other rooms. DrilDICE, leveraging the cost derived from $\pi_{bc}$, computes the corrected state distribution through the worst-case weighting, denoted as $w_{bc}^*$, as depicted in Figure 3-(c). Notably, states mispredicted by $\pi_{bc}$ or those with high probabilities for the dataset distribution tend to obtain relatively higher weight values, contributing more to loss optimization. Consequently, DrilDICE effectively trains the agent by correcting suboptimal behaviors.

## 4.4 D4RL Dataset with Covariate-Shifted Expert Demonstrations

### 4.4.1 Covariate Shift Scenarios

To investigate our approach also benefits distributionally robust training in more complex tasks, we conduct experiments on continuous control domains. In addition, we devise three practical problem settings that could possibly occur in the real-world and simulate those scenarios by restructuring the given dataset to simulate these dataset deviation scenarios (For experiments on standard scenarios, refer to Section E). We utilize `hopper-expert`, `walker2d-expert`, `halfcheetah-expert` datasets included in D4RL benchmark [9] as original datasets. We utilize the soft TV-distance defined in Section 3.3 as $f$-divergence for OptiDICE-BC and DrilDICE for these scenarios (For experiments with another choice of $f$-divergence, see Section E.4). See Section D for implementation details.

**Scenario 1: Rebalanced dataset**  From a practical perspective, a data collector may encounter scenarios in which the costs associated with taking an action or visiting a specific state exhibit substantial costs varying across different states or actions. In such scenarios, the data collected in states or actions with lower costs is likely to be observed, while those with higher costs tend to be underrepresented. To simulate these circumstances, we partition the state or action space and manipulate their mixture ratio to generate an imbalanced dataset, under assuming the costs of collecting data points in each group is significantly different, hence their proportion is shifted from the original dataset.

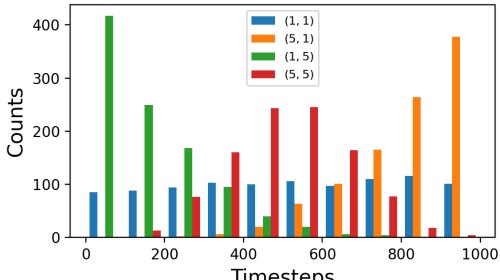
(a) Timesteps subsampled for Scenario 2.

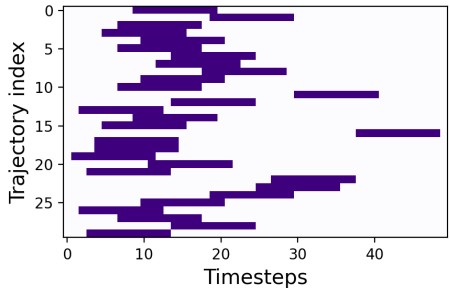
(b) Segments collected for Scenario 3.

Figure 4: Illustrative examples for generating datasets for Scenario 2 and 3.

| | Task | $(a, b)$ | BC | DemoDICE | AW-BC | DR-BC | OptiDICE-BC | DrilDICE (Ours) |
|---|---|---|---|---|---|---|---|---|
| Timestep dependency | hopper | (1, 1) | $28.9 \pm 3.8$ | $26.4 \pm 5.8$ | $18.0 \pm 3.1$ | $21.1 \pm 2.3$ | $22.8 \pm 3.9$ | $\mathbf{45.4 \pm 5.1}$ |
| | | (1, 5) | $31.0 \pm 0.9$ | $25.7 \pm 2.8$ | $24.9 \pm 1.6$ | $25.0 \pm 1.7$ | $19.3 \pm 1.2$ | $\mathbf{45.6 \pm 4.6}$ |
| | | (5, 1) | $26.8 \pm 7.1$ | $23.0 \pm 5.4$ | $23.2 \pm 4.6$ | $17.5 \pm 3.4$ | $25.7 \pm 6.0$ | $\mathbf{34.7 \pm 9.0}$ |
| | | (5, 5) | $27.7 \pm 6.7$ | $\mathbf{38.7 \pm 11.0}$ | $27.7 \pm 7.8$ | $23.2 \pm 6.3$ | $14.1 \pm 3.6$ | $25.6 \pm 6.0$ |
| | walker2d | (1, 1) | $29.0 \pm 5.3$ | $21.7 \pm 4.1$ | $27.6 \pm 3.7$ | $45.7 \pm 9.9$ | $6.1 \pm 1.0$ | $\mathbf{81.2 \pm 5.4}$ |
| | | (1, 5) | $61.5 \pm 5.2$ | $52.0 \pm 6.5$ | $34.7 \pm 5.9$ | $57.3 \pm 4.8$ | $17.6 \pm 2.7$ | $\mathbf{84.3 \pm 4.9}$ |
| | | (5, 1) | $8.1 \pm 0.7$ | $7.4 \pm 0.8$ | $8.8 \pm 2.2$ | $18.0 \pm 2.6$ | $4.4 \pm 0.5$ | $\mathbf{48.2 \pm 8.3}$ |
| | | (5, 5) | $6.7 \pm 1.2$ | $7.4 \pm 1.2$ | $10.6 \pm 3.3$ | $12.5 \pm 1.8$ | $5.5 \pm 0.7$ | $\mathbf{52.6 \pm 6.2}$ |
| | halfcheetah | (1, 1) | $33.7 \pm 3.0$ | $35.0 \pm 6.2$ | $33.4 \pm 3.5$ | $17.1 \pm 2.4$ | $9.9 \pm 3.6$ | $\mathbf{44.2 \pm 5.6}$ |
| | | (1, 5) | $72.7 \pm 2.6$ | $74.2 \pm 1.4$ | $71.1 \pm 2.2$ | $61.8 \pm 2.5$ | $24.4 \pm 3.3$ | $\mathbf{77.1 \pm 2.4}$ |
| | | (5, 1) | $2.4 \pm 0.5$ | $4.6 \pm 1.5$ | $4.2 \pm 0.4$ | $3.8 \pm 1.3$ | $1.3 \pm 1.1$ | $\mathbf{5.7 \pm 1.5}$ |
| | | (5, 5) | $2.0 \pm 0.9$ | $2.9 \pm 2.2$ | $0.9 \pm 0.7$ | $2.9 \pm 1.1$ | $-1.2 \pm 0.4$ | $\mathbf{5.5 \pm 0.8}$ |

Table 4: Performance comparison on Scenario 2 (time-dependently collected dataset). Each experiment is repeated 5 times, and the average normalized scores with their standard errors are reported. The highest mean performance scores are highlighted in bold.

In order to implement this scenario, we measure the distance between each state and the representative point of the states, then split the dataset into two groups $D_1$, $D_2$ based on the statistics of distances: $D_1$ comprises states close to the point, while $D_2$ is consists of states farther from the point. Subsequently, we resample datasets according to a predetermined proportion of $p(D_1)$ in $\{0.1, 0.5, 0.9\}$ by uniformly sampling different numbers of samples from each group.

**Scenario 2: Time-dependently collected dataset** One of well-known practical issues of covariate shift in supervised learning and time-series forecasting communities is dataset shift caused by seasonal or time-dependent variables [8, 18, 26]. In real-world scenarios, data collection agents affected by seasonality (e.g., temperature, humidity) can cause the data to be collected far from the expert agent's full demonstration. For instance, consider a sensor with the collecting frequency is sensitive to the temperature, making it prone to frequent breakdowns in the summer, and this sensor should collect expert data throughout the entire year. However, due to its frequent failures during the summer, the collected dataset will deviate from $d_E$.

Motivated by these scenarios, we simulate time-dependent covariate shift scenarios by subsampling timesteps of the trajectory in extensive manners. To model the timestep sampling distribution, we utilize Beta distribution $\mathrm{B}(a, b)$. Since the support of $\mathrm{B}(a, b)$ is $[0, 1]$, with multiplying with the maximum timestep and discretization, we can sample timesteps in various shapes of distributions by adjusting parameters $a, b$. We conduct experiments on four parameter combinations, $(a, b) \in \{(1, 1), (1, 5), (5, 1), (5, 5)\}$ and the timestep distribution notably varies as depicted in Figure 4a.

**Scenario 3: Segmented trajectory dataset** As similarly explored in [30], when decision-making horizons are extensively long, it is not feasible to gather multiple long trajectories keep tracking from initial states to ensure the stationarity of the expert dataset. Instead, it is more practical to gather shorter segments of expert demonstrations in such scenarios. We simulate this scenario by utilizing short segments of the pre-collected trajectories. To collect this, for each trajectory in the original dataset, we sample the starting timestep of the segment by using a subsampling method similar to Scenario 2. Then, we extract consequent segments with fixed-length timesteps and configure the segmented trajectory dataset similar to Figure 4b. In this scenario, we adjust the number of segments used for training to investigate the relationship between performance and the number of segments.

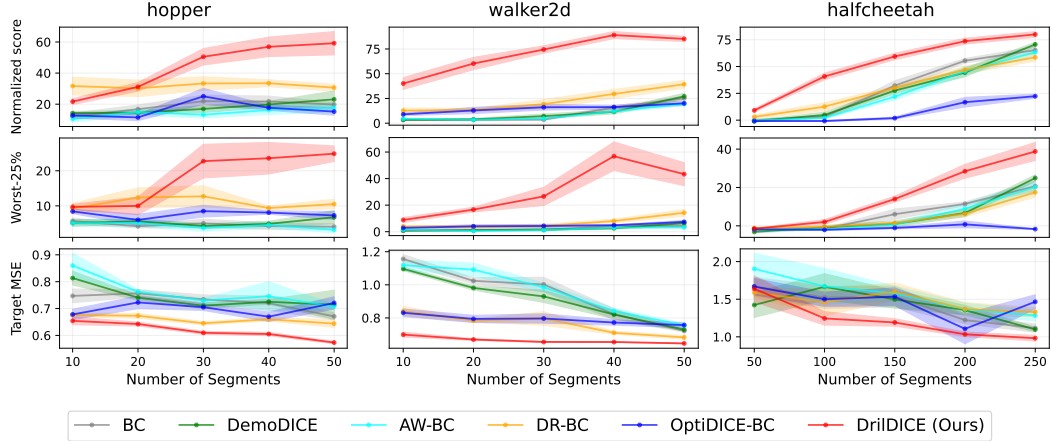

Figure 5: Performance comparison on Scenario 3 (segmented dataset) along the number of segments. The points and shaded areas indicate the means and standard errors measured over 5 repetitions.

### 4.4.2 Results

**Scenario 1**    Table 3 summarizes the performance comparison on the rebalanced datasets. From the table, we observe that DrilDICE overally outperforms other methods across different proportions and tasks and outperforms 14 out of 15 problem settings with signficant margins. DemoDICE, AW-BC and OptiDICE-BC fail to outperform BC more than half of the problem settings, which demonstrates the distribution matching and the best-case weighting are not robust to these shift scenarios. Moreover, we can conclude a robustness of DrilDICE is not gifted by side effects from the implementation of DICE algorithms at least this scenario. DR-BC, which does not incorporate Bellman flow constraints, also exhibits improved robustness, outperforming BC in 9 out of 15 problem settings. However, DrilDICE consistently surpasses DR-BC across all settings, illustrating that the inclusion of Bellman flow constraints is crucial for effectively addressing the covariate shift of our interest.

**Scenario 2**    As depicted in Table 4, DrilDICE shows robust overall performance, achieving the highest mean performance in 11 out of 12 settings. Remarkably, while all baseline methods, including DR-BC, fail to surpass BC more than half of the problem settings, only DrilDICE consistently demonstrates exceptional robustness. This emphasizes that the critical role of incorporating Bellman flow constraint to enhance robustness in scenarios involving this type of covariate shift.

**Scenario 3**    Figure 5 presents a performance comparison across three metrics while varying the number of segments. As illustrated, DrilDICE's normalized score consistently increases as the number of segments grows, and the target MSE exhibits a steady decreases maintaining levels that are lower or comparable to those of other approaches. The robust performance metrics also display consistent behavior, with an exception of `walker2d` task. Interestingly, we observed that a weak correlation between the target MSE and the episode return (see Figure E in the supplementary material).

In summary, by constraining the uncertainty set to a plausible set, DrilDICE effectively minimizes target MSE and enables BC to imitate the agent robust to various covariate shift scenarios.

## 5    Conclusions and Limitations

We propose DrilDICE, a offline IL approach that is robust to the covariate shift caused by the data distribution deviated from the stationary distribution of the expert. By optimizing the Bellman-flow-constrained worst-case objective, our approach effectively minimize the surrogate loss of the expected error w.r.t non-shifted target distribution. We also suggests an extensive set of practical covariate shift scenarios of our interest and empirically show that DrilDICE successfully imitate the expert robust to those shift scenarios. For a limitation, we don't consider uncertainty from transition shift or noisy demonstrations. We expect that extending our approach into such scenario will be beneficial on many practical scenarios, such as Sim2Real or transfer learning tasks. Moreover, we believe exploring a real-world application of our method is also an attractive extend of the future work.

## Acknowledgments and Disclosure of Funding

This work was supported by IITP grant funded by MSIT of Korea (No. RS-2020-II200940, No. RS-2022-II220311, No. RS-2019-II190075, No. RS-2019-II190079, No. RS-2024-00397310, No. RS-2024-00343989, No. RS-2024-00457882), IITP-ITRC (IITP-2024-RS-2024-00436857), NRF of Korea (No. RS-2024-00451162, No. RS-2023-00279680), BK21 Four project, KAIST-NAVER Hypercreative AI Center, and NSF AI4OPT AI Centre.

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

# Supplementary Material

## Contents

# A  Suboptimality Bounds with Arbitrary State Distributions

In this section, we present an example of BC where the performance guarantee fails when the data state distribution $d_D$ differs from expert stationary $d_E$. For simplicity, here we focus on the finite-horizon average state distribution, but this can be straightforwardly extended to the infinite-horizon discounted state distribution. Consider a finite-horizon setting with a horizon $H \geq 2$ and denote the state distribution $d_\pi^i$ of $\pi$ at timestep $i$. Define the $H$-horizon state distribution $d_\pi$ of $\pi$ as $d_\pi(s) := \frac{1}{H} \sum_{i=1}^{H} d_\pi^i(s)$ for all $s \in \mathcal{S}$. Denote the performance of $\pi$ as $J(\pi) := \sum_{t=0}^{H-1} \mathbb{E}_{d_\pi^t}[R(s_t, a_t)] = H\mathbb{E}_{d_\pi}[R(s,a)]$ where $R : \mathcal{S} \times \mathcal{A} \to [0,1]$ is a true reward function. Denote a surrogate loss function $\ell$, which measures how far the learner policy $\pi$ is from the expert policy for the state $s \in \mathcal{S}$. Here, consider 0-1 loss, i.e. $\ell(\pi(s), \pi_E(s)) = \mathbb{I}[\pi(s) \neq \pi_E(s)]$.

**Proposition 2.** *(Ross and Bagnell [21]) The suboptimality of an imitator policy $\hat{\pi}$ is bounded by*

$$J(\pi_E) - J(\hat{\pi}) \leq H^2 \mathbb{E}_{s \sim d_E}[\ell(\hat{\pi}(s), \pi_E(s))] \tag{11}$$

In our problem setting, we assume the lack of access to dataset sampled from $d_E(s)$, instead deal with dataset sampled from an arbitrary distribution $d_D(s)$. When applying BC approach to minimize $\mathbb{E}_{s \sim d_D}[\ell(\pi(s), \pi_E(s))]$, it is possible to construct an example where this expected loss fails to upper bound the policy suboptimality, $J(\pi_E) - J(\pi)$. To illustrate, we revisit [21] and consider an example that tightens the inequality (11). Assume $0 < \epsilon \leq 1/H, 0 < \delta < 1/H$.

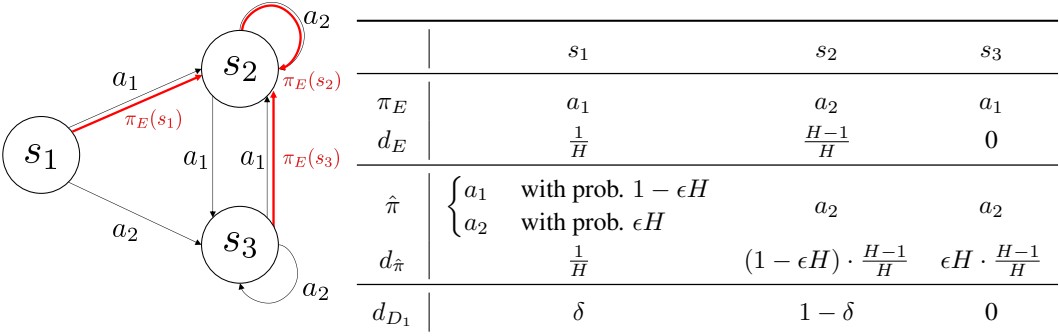

| | | $s_1$ | $s_2$ | $s_3$ |
|---|---|---|---|---|
| $\pi_E$ | | $a_1$ | $a_2$ | $a_1$ |
| $d_E$ | | $\frac{1}{H}$ | $\frac{H-1}{H}$ | $0$ |
| $\hat{\pi}$ | $\begin{cases} a_1 & \text{with prob. } 1 - \epsilon H \\ a_2 & \text{with prob. } \epsilon H \end{cases}$ | | $a_2$ | $a_2$ |
| $d_{\hat{\pi}}$ | | $\frac{1}{H}$ | $(1 - \epsilon H) \cdot \frac{H-1}{H}$ | $\epsilon H \cdot \frac{H-1}{H}$ |
| $d_{D_1}$ | | $\delta$ | $1 - \delta$ | $0$ |

Figure A: Example of [21].          Table A: Policies and their corresponding state distributions.

Consider an MDP illustrated in Figure A, where $\mathcal{S} = \{s_1, s_2, s_3\}, \mathcal{A} = \{a_1, a_2\}$, the transition $T$ is deterministic with $T(s_1, a_1) = s_2, T(s_1, a_2) = s_3, T(s_2, a_1) = s_3, T(s_2, a_2) = s_2, T(s_3, a_1) = s_2, T(s_3, a_2) = s_3$ and $s_1$ is the initial state. We define the expert policy $\pi_E$ and an imitator policy $\hat{\pi}$ and derive their corresponding $H$-horizon state distributions $d_E, d_{\hat{\pi}}$ in Table A. Define $R(s,a) = \mathbb{I}[a = \pi_E(s)]$ for all $s \in \mathcal{S}, a \in \mathcal{A}$. Then, $J(\pi_E), J(\hat{\pi})$ would be:

$$J(\pi_E) = H \sum_{s \in \mathcal{S}} d_E(s) \sum_{a \in \mathcal{A}} \pi_E(a|s) R(s,a) = H\left(\frac{1}{H} + \frac{H-1}{H}\right) = H$$

$$J(\hat{\pi}) = H \sum_{s \in \mathcal{S}} d_{\hat{\pi}}(s) \sum_{a \in \mathcal{A}} \hat{\pi}(a|s) R(s,a) = H\left(\frac{1}{H} \cdot (1 - \epsilon H) + \frac{H-1}{H} \cdot (1 - \epsilon H)\right) = H - \epsilon H^2$$

Hence, $J(\pi_E) - J(\hat{\pi}) = \epsilon H^2$. Note that $\mathbb{E}_{s \sim d_E}[\ell(\hat{\pi}, \pi_E)] = \frac{1}{H} \cdot \epsilon H + \frac{H-1}{H} \cdot 0 + 0 \cdot 1 = \epsilon$, therefore the both left-hand and right-hand sides of the equality (11) are equal to $\epsilon H^2$.

However, considering the expect loss with respect to $d_{D_1}$, where $d_{D_1}(s_1) = \delta, d_{D_1}(s_2) = 1 - \delta, d_{D_1}(s_3) = 0$ as described in Table A, we can calculate $\mathbb{E}_{s \sim d_{D_1}}[\ell(\hat{\pi}, \pi_E)] = \delta \epsilon H$. Then,

$$J(\pi_E) - J(\hat{\pi}) = \epsilon H^2 > \delta \epsilon H^3 = H^2 \mathbb{E}_{s \sim d_{D_1}}[\ell(\hat{\pi}, \pi_E)]$$

where $\delta < 1/H$. As $\delta$ decreases, the right-hand side also decreases, allowing it to be made arbitrarily small regardless of the left-hand side. This implies that there exist a scenario in which minimizing BC loss with an arbitrary data state distribution does not guarantee a reduction in the performance gap between the expert and the imitator.

Although straightforward, we emphasize that distributionally robust optimization (DRO) approaches can minimize the suboptimality under the realizability assumption, i.e, $d_E \in \mathcal{Q}$.

**Remark 1.** *Given an uncertainty set $\mathcal{Q} \subseteq \Delta(\mathcal{S})$ with $d_E \in \mathcal{Q}$, we can guarantee that $J(\pi_E) - J(\pi) \leq H^2 \sup_{d \in \mathcal{Q}} \mathbb{E}_{s \sim d}[l(\pi(s), \pi_E(s))]$. Furthermore, given $\mathcal{Q}_1 \subseteq \mathcal{Q}_2 \subseteq \Delta(\mathcal{S})$ with $d_E \in \mathcal{Q}_1$, the upper-bound of $\mathcal{Q}_1$ would be tighter than $\mathcal{Q}_2$, i.e.*

$$J(\pi_E) - J(\pi) \leq H^2 \sup_{d \in \mathcal{Q}_1} \mathbb{E}_{s \sim d}[l(\pi(s), \pi_E(s))] \leq H^2 \sup_{d \in \mathcal{Q}_2} \mathbb{E}_{s \sim d}[l(\pi(s), \pi_E(s))]$$

Let $\mathcal{Q}_\rho := \{d \in \Delta(\mathcal{S}) : D_{\text{TV}}(d, d_D) \leq \rho\}$ and $\mathcal{Q}_\rho^{\text{DrilDICE}} = \{d \in \Delta(\mathcal{S}) : D_{\text{TV}}(d, d_D) \leq \rho, d$ satisfies Bellman flow constraints$\}$. Since $d_E$ satisfies Bellman flow constraints, if $d_E \in \mathcal{Q}_\rho$, then $d_E \in \mathcal{Q}_\rho^{\text{DrilDICE}}$ with $\mathcal{Q}_\rho^{\text{DrilDICE}} \subseteq \mathcal{Q}_\rho$. Then, the uncertainty set of $\mathcal{Q}_\rho^{\text{DrilDICE}}$ provides a more tighter upper-bound for the suboptimality compared to that of $\mathcal{Q}_\rho$.

# B  Summary of $f$-Divergence Choices

Define
$$f_{\text{Soft-}\chi^2}(x) := \begin{cases} x \log x - x + 1 & \text{if } 0 < x < 1 \\ (x-1)^2 & \text{if } x \geq 1 \end{cases} \text{ and } (f'_{\text{Soft-}\chi^2})^{-1}(y) := \begin{cases} \exp(y) & \text{if } y < 0 \\ y + 1 & \text{if } y \geq 0 \end{cases},$$
$\text{ReLU}(x) := \max(0, x), \text{ELU}(x) := \begin{cases} \exp(x) - 1 & \text{if } x < 0 \\ x & \text{if } x \geq 0 \end{cases}$. Then, $f$-divergence choices can be summarized in Table B.

| **Divergence** | $f(x)$ | $(f')^{-1}(y)$ | $w^*_{\pi,\nu}$ |
|---|---|---|---|
| KL Divergence | $x \log x$ | $\exp(y-1)$ | $\exp\left(\frac{e_{\pi,\nu}(s,a,s')}{\alpha} - 1\right)$ |
| $\chi^2$-Divergence | $\frac{1}{2}(x-1)^2$ | $(y+1)$ | $\text{ReLU}\left(\frac{e_{\pi,\nu}(s,a,s')}{\alpha} + 1\right)$ |
| Soft $\chi^2$-Divergence | $f_{\text{Soft-}\chi^2}(x)$ | $(f')^{-1}_{\text{Soft-}\chi^2}(y)$ | $\text{ELU}\left(\frac{e_{\pi,\nu}(s,a,s')}{\alpha} + 1\right)$ |
| TV-Distance | $\frac{1}{2}\lvert x-1 \rvert$ | - | - |
| Soft TV-Distance | $\frac{1}{2}\log(\cosh(x-1))$ | $\tanh^{-1}(2y)$ | $\text{ReLU}\left(\tanh^{-1}\left(\frac{2e_{\pi,\nu}(s,a,s')}{\alpha}\right) + 1\right)$ |

Table B: Summary of $f$-divergences and their associated functions.

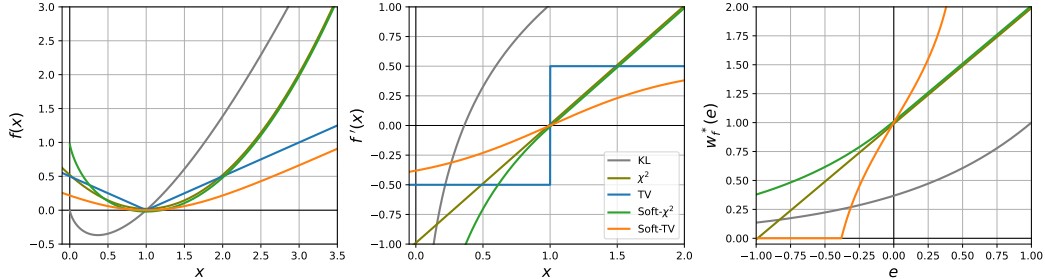

Figure B: Visualization of $f$ functions and derivatives $f'$ corresponding to $f$-divergence choices. We also visualize corresponding $w^*_f(e) = \max((f')^{-1}(e), 0)$, which is a closed form solution of the inner maximization in Eq. 7.

# C    Experimental Settings for Four Rooms Environment

## C.1    Marginal probability of the original dataset

| Rooms | Room 1 | Room 2 | Room 3 | Room 4 |
|---|---|---|---|---|
| $D_E(u)$ | 0.342 | 0.291 | 0.089 | 0.278 |
| **Actions** | **"UP"** | **"DOWN"** | **"LEFT"** | **"RIGHT"** |
| $D_E(u)$ | 0.080 | 0.453 | 0.016 | 0.451 |

Table C: Marginal probability of $D_E$ over rooms and actions.

## C.2    Implementation details

- The number of transitions: $1000 + |\mathcal{S}|^4$
- RBF feature bandwidth: 10
- The number of representative points for RBF feature : $5 \times 5$
- $C_\pi$: 1-0 loss

## C.3    Hyperparameters

- $\alpha \in \{100, 50, 20, 10, 5, 2, 1, 0.5, 0.2, 0.1\}$ (for OptiDICE-BC, DrilDICE)
- $\rho \in \{100, 50, 20, 10, 5, 2, 1, 0.5, 0.2, 0.1\}$ (for DR-BC)
- Logit margin maximum: $\log(100)$

# D    Experimental Settings for D4RL Benchmark

## D.1    Implementation details

- Since the number of $s_0$ is much smaller than $s$, we follow heuristics to estimate $\mathbb{E}_{s \sim d_D}[\nu(s)]$ instead of $\mathbb{E}_{s \sim \rho_0}[\nu(s)]$ in the objective (8).
- In Scenario 3, to determine a starting timestep of each segment, we sample timesteps by using Geometric distribution with $p = 5 \times 10^{-3}$.
- $C_\pi$: Mean Squared Error (MSE) loss

## D.2    Hyperparameters

| Hyperparameter | BC | DemoDICE | AW-BC | DR-BC | OptiDICE-BC | DrilDICE (Ours) |
|---|---|---|---|---|---|---|
| Policy distribution | | | | Tanh Normal | | |
| Batch size | | | | 512 | | |
| Policy learning rate | | | | $3 \times 10^{-5}$ | | |
| hidden units | | | | $[256, 256]$ | | |
| Training iteration | | | | 500K | | |
| $\alpha$ | - | - | - | - | $[10^{-1}, 10^{-2}, 10^{-3}, 10^{-4}]$ | |
| $\rho$ | - | - | - | $[10^{-1}, 10^{-2}, 10^{-3}, 10^{-4}]$ | - | - |
| $\nu$ (or $w$) learning rate | - | | | $3 \times 10^{-5}$ | | |

Table D: Summary of hyperparameters used in D4RL benchmark experiments.

---

[4]To prevent support mismatch, one data point for each $s \in \mathcal{S}$ is added to dataset.

| Problem setting | hopper | walker2d | halfcheetah |
|---|---|---|---|
| Scenario 1 | 100 | 100 | 250 |
| Scenario 2 | 100 | 100 | 250 |
| Scenario 3 | [10,20,30,40,50] | [10,20,30,40,50] | [50,100,150,200,250] |

Table E: Number of sub-trajectories used in each scenario. Each sub-trajectory consists of one initial transition and 50 subsampled transitions from a complete trajectory. To enhance dataset support, one complete trajectory is appended for each specified count of sub-trajectories.

# E   Additional Experimental Results

## E.1   Performance comparison on complete trajectories

To compare performance of IL methods on standard scenarios, we have conducted additional experiments using complete expert trajectories using D4RL expert-v2 dataset varying the number of trajectories in {1, 5, 10, 50}. The results are detailed in Figure C. As demonstrated, DrilDICE can deal with sampling errors of small datasets, showing a superior data efficiency compared to other methods.

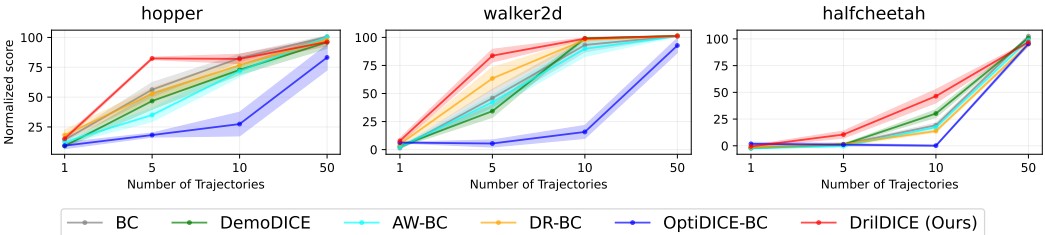

Figure C: Performance comparison on complete trajectory scenarios. The points/shaded areas indicate the means/standard errors measured over 5 repetitions.

## E.2   Performance comparison on lower quality segments

Despite our primary focus on expert-quality datasets, we conducted experiments with additional datasets on Scenario 3 (segment datasets) to ensure consistency across different datasets. Rather than employing the original dataset, D4RL expert-v2, we use D4RL medium-v2 quality demonstrations as our imitation standard for comparison. The results are depicted in Figure D. The results indicate that both DrilDICE and DR-BC demonstrate competitive imitation performance compared to other baselines, with a consistent decrease in target MSE as the number of segments increases.

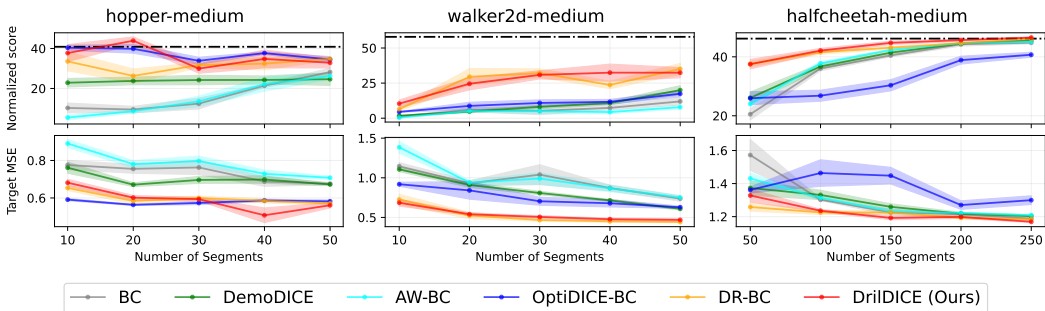

Figure D: Performance comparison on Scenario 3 (segmented dataset) with D4RL medium-v2 datasets. Each black dashed-dot line expresses the averaged normalized score of the used dataset. The points/shaded areas indicate the means/standard errors measured over 5 repetitions.

## E.3  Weak correlation between target MSE and normalized score

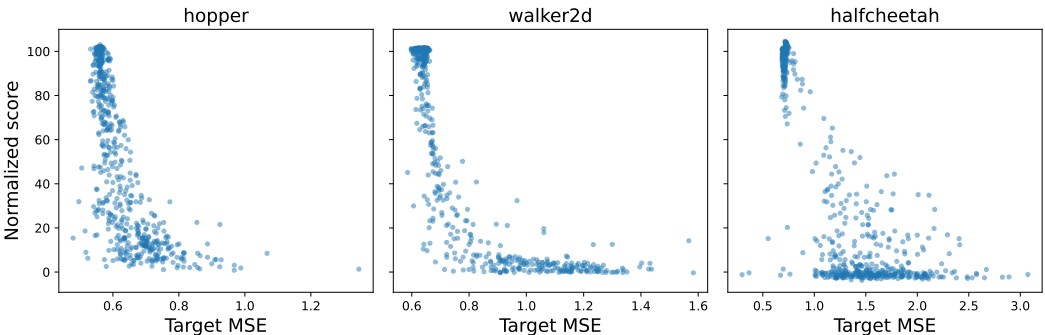

Figure E: Correlation between the target MSE and normalized scores.

## E.4  Performance comparison with a different $f$-divergence choice

To evaluate the effect of $f$-divergence choice on imitation performance, we additionally evaluate OptiDICE-BC and DrilDICE with the soft $\chi^2$-divergence, as originally utilized in [16], across all considered scenarios. The results for scenarios 1-3 are presented in Table F, G and Figure F respectively. These results indicate that the choice of the soft TV-distance significantly enhances the performance of DrilDICE compared to using soft-$\chi^2$. To explain this performance gain, we hypothesize that the soft-TV distance provides more discriminative weighting of samples based on their long-term policy errors. As shown in Figure B, conventional $f$-divergences (e.g. KL, soft-$\chi^2$, ...) make less pronounced to the long-term policy error, yet the soft-TV distance responds sensitively to changes in $e$, resulting in a more pronounced weight $w$. This enables BC loss to more selectively focus on critical samples with significant magnitude of long-term policy errors, thereby effectively enhancing performance, akin to the benefits observed in Sparse Q-Learning [29].

|  | Task | $p(D_1)$ | BC | DR-BC (TV) | OptiDICE-BC (Soft-$\chi^2$) | OptiDICE-BC (Soft-TV) | DrilDICE (Soft-$\chi^2$) | DrilDICE (Soft-TV) |
|---|---|---|---|---|---|---|---|---|
| Rebalanced by state | hopper | 0.1 | 24.7 ± 4.2 | 27.0 ± 4.3 | 35.3 ± 3.2 | 12.7 ± 1.3 | **58.9 ± 4.3** | 52.2 ± 5.6 |
| | | 0.5 | 35.4 ± 4.1 | 36.7 ± 3.6 | 24.4 ± 5.1 | 8.6 ± 2.0 | 60.9 ± 7.6 | **67.1 ± 8.2** |
| | | 0.9 | 11.2 ± 2.5 | 27.4 ± 4.9 | 17.3 ± 2.1 | 10.4 ± 1.8 | 28.3 ± 3.0 | **36.4 ± 6.1** |
| | walker2d | 0.1 | 18.9 ± 4.1 | 14.7 ± 3.3 | 5.9 ± 0.6 | 4.9 ± 1.1 | 31.9 ± 5.9 | **51.6 ± 8.2** |
| | | 0.5 | 22.9 ± 3.1 | 8.1 ± 0.4 | 11.1 ± 1.4 | 8.1 ± 0.4 | 30.5 ± 3.5 | **73.7 ± 5.4** |
| | | 0.9 | 30.4 ± 7.1 | 7.7 ± 0.5 | 17.5 ± 2.6 | 7.7 ± 0.5 | 43.3 ± 4.5 | **77.6 ± 5.5** |
| | halfcheetah | 0.1 | 49.3 ± 5.2 | 32.9 ± 3.8 | 33.4 ± 4.7 | 7.1 ± 1.5 | 44.8 ± 5.2 | **52.5 ± 3.6** |
| | | 0.5 | 38.0 ± 3.1 | 26.2 ± 4.9 | 33.4 ± 3.0 | 6.1 ± 1.2 | 41.1 ± 3.5 | **55.0 ± 3.3** |
| | | 0.9 | 15.5 ± 3.1 | 9.0 ± 3.3 | 2.2 ± 1.2 | 1.0 ± 1.1 | 7.3 ± 1.6 | **22.3 ± 2.9** |
| Rebalanced by action | hopper | 0.1 | 29.7 ± 4.0 | 25.9 ± 2.5 | 28.4 ± 1.1 | 11.7 ± 2.1 | 42.3 ± 6.4 | **56.6 ± 11.9** |
| | | 0.5 | 26.4 ± 4.9 | 35.1 ± 5.4 | 30.0 ± 4.3 | 11.8 ± 1.1 | 53.4 ± 9.5 | **73.8 ± 3.6** |
| | | 0.9 | 30.5 ± 3.6 | 36.6 ± 2.3 | 38.9 ± 4.7 | 19.4 ± 2.8 | **63.1 ± 7.1** | 49.0 ± 12.3 |
| | walker2d | 0.1 | 23.6 ± 5.1 | 31.2 ± 4.2 | 12.1 ± 1.2 | 7.3 ± 0.5 | 40.7 ± 1.1 | **70.6 ± 3.2** |
| | | 0.5 | 32.3 ± 6.7 | 30.5 ± 3.9 | 16.4 ± 1.7 | 6.4 ± 1.0 | 47.9 ± 12.4 | **72.1 ± 8.7** |
| | | 0.9 | 16.9 ± 2.8 | 37.6 ± 9.0 | 15.6 ± 3.6 | 4.6 ± 1.0 | 43.7 ± 11.5 | **69.5 ± 8.5** |
| | halfcheetah | 0.1 | 41.9 ± 4.8 | 27.5 ± 1.0 | 26.6 ± 2.5 | 8.4 ± 3.4 | 32.8 ± 1.9 | **56.4 ± 4.6** |
| | | 0.5 | 45.8 ± 4.5 | 33.4 ± 6.5 | 45.5 ± 3.3 | 4.6 ± 0.8 | 48.1 ± 5.5 | **60.8 ± 1.6** |
| | | 0.9 | 25.9 ± 3.4 | 12.1 ± 2.0 | 4.3 ± 1.5 | 0.6 ± 0.7 | 9.6 ± 2.3 | **29.2 ± 4.6** |

Table F: Performance comparison on Scenario 1 (rebalanced dataset) including a different choice of $f$-divergence (soft $\chi^2$-divergence) for OptiDICE-BC and DrilDICE.

| | Task | $p(D_1)$ | BC | DR-BC (TV) | OptiDICE-BC (Soft-$\chi^2$) | OptiDICE-BC (Soft-TV) | DrilDICE (Soft-$\chi^2$) | DrilDICE (Soft-TV) |
|---|---|---|---|---|---|---|---|---|
| Timestep dependency | hopper | (1, 1) | $28.9 \pm 3.8$ | $21.1 \pm 2.3$ | $50.3 \pm 6.6$ | $22.8 \pm 3.9$ | $\mathbf{54.8 \pm 7.7}$ | $45.4 \pm 5.1$ |
| | | (1, 5) | $31.0 \pm 0.9$ | $25.0 \pm 1.7$ | $39.5 \pm 4.0$ | $19.3 \pm 1.2$ | $37.2 \pm 7.9$ | $\mathbf{45.6 \pm 4.6}$ |
| | | (5, 1) | $26.8 \pm 7.1$ | $17.5 \pm 3.4$ | $\mathbf{48.4 \pm 13.0}$ | $25.7 \pm 6.0$ | $39.9 \pm 9.2$ | $34.7 \pm 9.0$ |
| | | (5, 5) | $27.7 \pm 6.7$ | $23.2 \pm 6.3$ | $32.5 \pm 10.8$ | $14.1 \pm 3.6$ | $\mathbf{40.2 \pm 7.4}$ | $25.6 \pm 6.0$ |
| | walker2d | (1, 1) | $29.0 \pm 5.3$ | $45.7 \pm 9.9$ | $17.4 \pm 3.0$ | $6.1 \pm 1.0$ | $51.9 \pm 5.3$ | $\mathbf{81.2 \pm 5.4}$ |
| | | (1, 5) | $61.5 \pm 5.2$ | $57.3 \pm 4.8$ | $37.3 \pm 5.7$ | $17.6 \pm 2.7$ | $64.5 \pm 7.9$ | $\mathbf{84.3 \pm 4.9}$ |
| | | (5, 1) | $8.1 \pm 0.7$ | $18.0 \pm 2.6$ | $4.4 \pm 0.9$ | $4.4 \pm 0.5$ | $23.3 \pm 3.4$ | $\mathbf{48.2 \pm 8.3}$ |
| | | (5, 5) | $6.7 \pm 1.2$ | $12.5 \pm 1.8$ | $8.5 \pm 2.3$ | $5.5 \pm 0.7$ | $14.4 \pm 3.2$ | $\mathbf{52.6 \pm 6.2}$ |
| | halfcheetah | (1, 1) | $33.7 \pm 3.0$ | $17.1 \pm 2.4$ | $33.7 \pm 5.5$ | $9.9 \pm 3.6$ | $33.4 \pm 3.7$ | $\mathbf{44.2 \pm 5.6}$ |
| | | (1, 5) | $72.7 \pm 2.6$ | $61.8 \pm 2.5$ | $52.9 \pm 4.0$ | $24.4 \pm 3.3$ | $69.6 \pm 3.9$ | $\mathbf{77.1 \pm 2.4}$ |
| | | (5, 1) | $2.4 \pm 0.5$ | $3.8 \pm 1.3$ | $2.5 \pm 1.1$ | $1.3 \pm 1.1$ | $4.0 \pm 1.2$ | $\mathbf{5.7 \pm 1.5}$ |
| | | (5, 5) | $2.0 \pm 0.9$ | $2.9 \pm 1.1$ | $1.6 \pm 1.3$ | $-1.2 \pm 0.4$ | $4.6 \pm 1.6$ | $\mathbf{5.5 \pm 0.8}$ |

Table G: Performance comparison on Scenario 2 (time-dependently collected dataset) including a different choice of $f$-divergence (soft $\chi^2$-divergence) for OptiDICE-BC and DrilDICE.

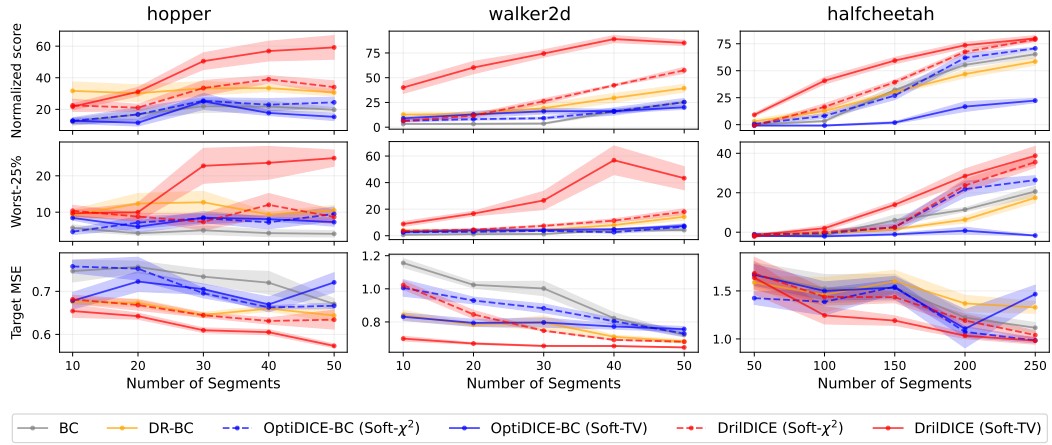

Figure F: Performance comparison on Scenario 3 (segmented dataset) including a different choice of $f$-divergence (soft $\chi^2$-divergence) for OptiDICE-BC and DrilDICE.

# F  Licenses

For all experiments in Four Rooms environment, we use CVXPY and convex optimization solver MOSEK with academic licenses. We also use Mujoco free licenses. Our code has been developed upon MIT licenses.

# G  Computation Resources

We used Google cloud computing engine with 100 `c2-standard-4` instances that have the following system specification:

- Series : C2
- Family : `compute-optimized`
- vCPU : 4
- Memory : 16 GB
- CPU Manufacturer : Intel
- CPU Platform : Intel Cascade Lake

- CPU Base Frequency : 3.1 GHz
- CPU Turbo Frequency : 3.8 GHz
- CPU Max. Turbo Frequency : 3.9 GHz
- Network Bandwidth : 10 Gbps
- Max. Disk Size : 257 TB
- Max. Number of Disks : 128

# H  Broader Impacts

Since we study distributionally robust learning, our approach can be used in fairness or bias reduction. As we discussed in experiment section, our algorithm is also beneficial when the training agent in the cost-constrained environment. Further, we expect our research contributes to build more trustworthy and robust AI systems.

