# OpenReview forum: "Mitigating Covariate Shift in Behavioral Cloning via Robust Stationary Distribution Correction"
_NeurIPS.cc/2024/Conference — NeurIPS 2024 poster_

### Official Review · Reviewer_k6J7 · 2024-07-01

**Soundness:** 3
**Presentation:** 3
**Contribution:** 3
**Rating:** 6
**Confidence:** 2

**Summary:**

This paper is motivated by the observation that BC is well-known to be vulnerable to the covariate shift resulting from the mismatch between the state distributions induced by the learned policy and the data collection policy. To solve this problem, the authors formulate a robust BC training objective and employ a stationary distribution correction ratio estimation (DICE) to derive a feasible solution. They evaluate the effectiveness of our method through an extensive set of experiments covering diverse covariate shift scenarios.

**Strengths:**

1.	This paper is well-written and well-organized.
2. The proposed method seems promising.
3.	A lot of covariate shift scenarios are investigated in the experiments, including discrete navigation tasks and mujoco control tasks.

**Weaknesses:**

1.	The experimental results do not fully support the advantages over baselines.

**Questions:**

1.	The advantage over OptiDICE-BC seems marginal in Table 2 and Table 4. What could be the reason?
2.	What does “marginal room (or action) distribution of dataset” mean in line 189?
3.	For mujoco experiments, the authors only use the expert split. How does the method perform in other splits that have lower quality?

**Limitations:**

One limitation mentioned by the authors in the last section is that: don’t consider uncertainty from transition shifts or noisy demonstrations.

---

> ### Author Rebuttal · Authors · 2024-08-07
>
> Thank you for your insightful feedback.
>
> **1. Addressing Concerns on Marginal Performance Improvements Compared to Baselines**
>
> In response, we have updated our main experiments to include the DRO baseline DR-BC [18] and variation of $f$-divergence for our approach accordingly: inspired by the choice of $f$ in DR-BC, we introduced **the soft-TV distance**, where the $f$ function is a log-cosh function and its derivative is the tanh function. [A] This enables us to obtain a closed form solution of $w$ by using Proposition 1.
>
> Based on this, we have expanded our analysis by incorporating three additional methods: **DR-BC** for all scenarios, **OptiDICE-BC (with soft-TV)**, and **DrilDICE (with soft-TV)** for Mujoco scenarios.
> The expanded results for Four Rooms environment and Scenario 1,2 can be found in **Table A, B, C** in the following comment, and the result for Scenario 3 can be found in **Figure B** in the PDF.
>
> The results consistently show that DrilDICE with the soft-TV outperforms baselines including DR-BC across most scenarios. Notably, DrilDICE with soft-TV, utilizing $f$-divergence similar to the one used in DR-BC, consistently outperforms DR-BC in most evaluation scenarios. We attribute this performance improvement to the remaining key difference: the inclusion of a Bellman flow constraint, which is a key contribution of our work.
>
> The results also demonstrate that the choice of the soft-TV significantly enhances the performance of DrilDICE compared to using soft-$\chi^2$. To explain this performance gain, we hypothesize that the soft-TV distance provides more discriminative weighting $w$ of samples based on their long-term policy errors. As shown in **Figure A** in the PDF, conventional $f$-divergences (e.g. KL, soft-$\chi^2$, …) make $w$ less pronounced to the long-term policy error $e$, yet the soft-TV distance responds sensitively to changes in $e$, resulting in a more pronounced $w$. This enables BC loss to more selectively focus on critical samples with significant magnitude of long-term policy errors, thereby effectively enhancing performance, akin to the benefits observed in Sparse Q-Learning [C].
>
> In summary, the choice of the $f$-divergence for DrilDICE is critical for performance. The use of soft-TV distance enhanced DrilDICE's performance across most problem settings, directly addressing the reviewer's concerns regarding its comparative performance. We hope these updates can address your concerns about the initial marginal improvements.
>
> [A] Saleh et al., "Statistical Properties of the log-cosh Loss Function used in Machine Learning." arXiv preprint (2022).
>
> [B] Xu et al., “Offline RL with No OOD Actions: In-sample Learning via Implicit Value Regularization”, ICLR 2023.
>
>
> **2. Clarification on the Terminology "Marginal Room (or Action) Distribution of Dataset"**
>
> In line 189, the term "marginal room (or action) distribution of dataset" specifically refers to $p(u)$,
> the proportion of transitions containing target factors (e.g. room visitation, action) within the manipulated dataset. To illustrate, these factors in our experiment include (1) the room visitation and (2) the action of the transition. By intentionally manipulating frequencies of these factors, we designed covariate shift scenarios.
>
> For example, in the Room 1 manipulation scenario, we initially split the expert dataset into two subsets $D_A,D_B$ based on whether each transition’s state was associated with Room 1 or not. We then subsampled transitions from subsets $D_A, D_B$ using the predetermined proportion $p(u), 1-p(u)$, respectively and combined them to construct the shifted dataset $D_i$.
>
> Here, the term "a marginal room (or action) distribution" refers to these proportions $p(u)$, representing how frequently transitions with a target factor (e.g. room visiting, action) are sampled relative to others in the dataset. We will revise this into a more clear term such as “a proportion of transitions containing a target factor” and spend our effort to improve presentations.
>
>
> **3. Performance Comparison with Lower Quality Segments**
>
> Despite our primary focus on expert-quality datasets, we conducted experiments with additional datasets on Scenario 3 (segment datasets) to ensure consistency across different datasets.
> Rather than employing the D4RL `expert-v2`, we use `medium-v2` quality demonstrations as our imitation standard for comparison. The results are depicted in **Figure D** of the PDF.
>
> Given that relying solely on the normalized scores may not accurately reflect the fidelity of imitation for a medium-quality policy, we primarily measured the target MSE for our comparisons. The results indicate that both DrilDICE and DR-BC demonstrate competitive imitation performance compared to other baselines, with a consistent decrease in target MSE as the number of segments increases.
>
>
> Thank you once more for your valuable feedback. If there are any more questions or concerns, please feel free to respond, and we will address them quickly.

---

> ### Author Response · Authors · 2024-08-07
> **Additional Experimental Results**
>
> ## **Four Rooms Environment**
>
> | Scenario | BC | OptiDICE-BC | DR-BC | DrilDICE (Ours) |
> | --- | --- | --- | --- | --- |
> | Room 1 | 90.84 ± 0.69 | 94.30 ± 0.41 | 91.38 ± 0.73 | **95.04 ± 0.48** |
> | Room 2 | 89.16 ± 1.07 | 94.06 ± 0.65 | 89.28 ± 1.08 | **94.44 ± 0.62** |
> | Room 3 | 88.50 ± 1.27 | 94.20 ± 0.98 | 88.70 ± 1.26 | **95.04 ± 0.86** |
> | Room 4 | 90.94 ± 0.75 | 94.26 ± 0.54 | 90.94 ± 0.75 | **94.92 ± 0.37** |
> | Action UP | 84.96 ± 1.33 | 92.06 ± 0.69 | 84.96 ± 1.33 | **93.22 ± 0.61** |
> | Action DOWN | 89.96 ± 0.96 | 93.62 ± 0.62 | 89.96 ± 0.96 | **94.60 ± 0.39** |
> | Action LEFT | 90.18 ± 1.11 | 91.86 ± 1.03 | 90.18 ± 1.11 | **92.62 ± 0.95** |
> | Action RIGHT | 93.04 ± 0.63 | 94.46 ± 0.44 | 93.44 ± 0.60 | **94.52 ± 0.44** |
>
> **Table A.** Expanded performance comparison of normalized scores on Four Rooms environment. (corresponds to Table 2)
>
> ## **Scenario 1: Rebalanced Dataset**
>
> | Scenario | Task | $p(D_1)$ | BC | OptiDICE-BC (Soft-$\chi^2$) | DrilDICE (Soft-$\chi^2$) | DR-BC | OptiDICE-BC (Soft-TV) | DrilDICE (Soft-TV) |
> | --- | --- | --- | --- | --- | --- | --- | --- | --- |
> | Rebalanced by state | hopper | 0.1 | 24.65 ± 4.15 | 35.26 ± 3.19 | **58.92 ± 4.30** | 27.02 ± 4.31 | 12.72 ± 1.27 | 52.22 ± 5.57 |
> |  |  | 0.5 | 35.38 ± 4.08 | 24.41 ± 5.12 | 60.91 ± 7.61 | 36.71 ± 3.56 | 8.61 ± 1.97 | **67.12 ± 8.18** |
> |  |  | 0.9 | 11.23 ± 2.49 | 17.26 ± 2.10 | 28.29 ± 3.03 | 27.37 ± 4.89 | 10.44 ± 1.76 | **36.39 ± 6.10** |
> |  | walker2d | 0.1 | 18.85 ± 4.05 | 5.91 ± 0.59 | 31.92 ± 5.94 | 14.69 ± 3.26 | 4.91 ± 1.08 | **51.55 ± 8.16** |
> |  |  | 0.5 | 22.91 ± 3.13 | 11.12 ± 1.36 | 30.53 ± 3.47 | 45.08 ± 9.98 | 8.09 ± 0.42 | **73.74 ± 5.37** |
> |  |  | 0.9 | 30.42 ± 7.05 | 17.51 ± 2.61 | 43.31 ± 4.54 | 46.04 ± 8.35 | 7.69 ± 0.45 | **77.60 ± 5.45** |
> |  | halfcheetah | 0.1 | 49.31 ± 5.16 | 33.42 ± 4.70 | 44.79 ± 5.16 | 32.85 ± 3.82 | 7.05 ± 1.47 | **52.45 ± 3.62** |
> |  |  | 0.5 | 37.98 ± 3.07 | 33.36 ± 3.04 | 41.13 ± 3.53 | 26.16 ± 4.91 | 6.14 ± 1.21 | **55.04 ± 3.27** |
> |  |  | 0.9 | 15.54 ± 3.06 | 2.21 ± 1.16 | 7.28 ± 1.59 | 8.96 ± 3.26 | 1.02 ± 1.12 | **22.28 ± 2.88** |
> | Rebalanced by action | hopper | 0.1 | 29.71 ± 4.00 | 28.37 ± 1.11 | 42.29 ± 6.39 | 25.92 ± 2.45 | 11.71 ± 2.12 | **56.60 ± 11.90** |
> |  |  | 0.5 | 26.35 ± 4.88 | 30.03 ± 4.23 | 53.37 ± 9.50 | 35.13 ± 5.41 | 11.79 ± 1.14 | **73.80 ± 3.63** |
> |  |  | 0.9 | 30.50 ± 3.60 | 38.92 ± 4.78 | **63.14 ± 7.12** | 36.56 ± 2.29 | 19.42 ± 2.82 | 48.99 ± 12.27 |
> |  | walker2d | 0.1 | 23.61 ± 5.10 | 12.06 ± 1.20 | 40.72 ± 1.13 | 31.18 ± 4.23 | 7.27 ± 0.46 | **70.60 ± 3.21** |
> |  |  | 0.5 | 32.29 ± 6.74 | 16.40 ± 1.70 | 47.93 ± 12.36 | 30.52 ± 3.89 | 6.37 ± 0.97 | **72.09 ± 8.70** |
> |  |  | 0.9 | 16.87 ± 2.80 | 15.64 ± 3.57 | 43.68 ± 11.50 | 37.55 ± 8.97 | 4.60 ± 0.97 | **69.51 ± 8.54** |
> |  | halfcheetah | 0.1 | 41.91 ± 4.80 | 26.62 ± 2.54 | 32.76 ± 1.87 | 27.50 ± 1.01 | 8.41 ± 3.38 | **56.42 ± 4.57** |
> |  |  | 0.5 | 45.80 ± 4.45 | 45.52 ± 3.24 | 48.08 ± 5.50 | 33.39 ± 6.52 | 4.64 ± 0.84 | **60.81 ± 1.56** |
> |  |  | 0.9 | 25.91 ± 3.35 | 4.28 ± 1.52 | 9.57 ± 2.34 | 12.08 ± 2.01 | 0.59 ± 0.68 | **29.19 ± 4.58** |
>
> **Table B.** Expanded performance comparison on Scenario 1 (rebalanced dataset). (corresponds to Table 3)
>
>
> ## **Scenario 2: Time-dependently Subsampled Dataset**
>
> | Task | (a, b) | BC | OptiDICE-BC (Soft-$\chi^2$) | DrilDICE (Soft-$\chi^2$) | DR-BC | OptiDICE-BC (Soft-TV) | DrilDICE  (Soft-TV) |
> | --- | ------- | --- | --- | --- | --- | --- | --- |
> | hopper | (1, 1) | 28.89 ± 3.77 | 50.33 ± 6.60 | **54.83 ± 7.66** | 21.10 ± 2.26 | 22.77 ± 3.94 | 45.44 ± 5.11 |
> |  | (1, 5) | 31.03 ± 0.90 | 39.54 ± 4.02 | 37.18 ± 7.92 | 25.00 ± 1.66 | 19.25 ± 1.21 | **45.60 ± 4.63** |
> |  | (5, 1) | 26.75 ± 7.12 | **48.40 ± 12.98** | 39.91 ± 9.20 | 17.51 ± 3.38 | 25.68 ± 6.01 | 34.71 ± 9.00 |
> |  | (5, 5) | 27.65 ± 6.71 | 32.46 ± 10.79 | **40.24 ± 7.41** | 23.20 ± 6.32 | 14.12 ± 3.59 | 25.61 ± 6.03 |
> | walker2d | (1, 1) | 28.95 ± 5.34 | 17.42 ± 3.00 | 51.85 ± 5.30 | 45.66 ± 9.92 | 6.13 ± 1.03 | **81.21 ± 5.40**|
> |  | (1, 5) | 61.48 ± 5.19 | 37.25 ± 5.66 | 64.46 ± 7.92 | 57.29 ± 4.79 | 17.55 ± 2.65 | **84.28 ± 4.89** |
> |  | (5, 1) | 8.13 ± 0.72 | 4.43 ± 0.91 | 23.31 ± 3.44 | 17.97 ± 2.58 | 4.37 ± 0.52 | **48.23 ± 8.30** |
> |  | (5, 5) | 6.65 ± 1.20 | 8.50 ± 2.27 | 14.40 ± 3.21 | 12.45 ± 1.84 | 5.54 ± 0.68 | **52.57 ± 6.24** |
> | halfcheetah | (1, 1) | 33.74 ± 2.99 | 33.65 ± 5.49 | 33.43 ± 3.73 | 17.09 ± 2.43 | 9.93 ± 3.56 | **44.17 ± 5.62** |
> |  | (1, 5) | 72.72 ± 2.60 | 52.94 ± 3.98 | 69.63 ± 3.86 | 61.81 ± 2.50 | 24.42 ± 3.27 | **77.12 ± 2.42** |
> |  | (5, 1) | 2.35 ± 0.51 | 2.46 ± 1.05 | 3.97 ± 1.18 | 3.81 ± 1.26 | 1.29 ± 1.13 | **5.68 ± 1.50** |
> |  | (5, 5) | 2.01 ± 0.91 | 1.61 ± 1.30 | 4.61 ± 1.55 | 2.85 ± 1.05 | -1.19 ± 0.37 | **5.50 ± 0.83** |
>
> **Table C.** Expanded performance comparison on Scenario 2 (time-dependently collected dataset). (corresponds to Table 4)

---

> > ### Comment · Reviewer_k6J7 · 2024-08-12
> > **Reply to authors**
> >
> > Thanks to the authors for adding new experimental results to address my concerns. The new results provide new evidence and support for the proposed method. I don't have further questions and I raised my score to 6 to lean to accept this paper.

---

> > > ### Author Response · Authors · 2024-08-12
> > >
> > > We're glad to hear that your concerns have been addressed, and we sincerely appreciate your positive review!
> > >
> > > We will incorporate all the findings discussed with you into the revised manuscript.

---

> ### Author Response · Authors · 2024-08-12
> **Dear Reviewer k6J7**
>
> We respectfully remind you that less than 48 hours remain in our discussion period. We are dedicated to addressing any remaining concerns you may have. In short, we addressed your key concerns as follows:
>
> - **Concerns on Marignal Improvements**:
>     - We found that performance of DrilDICE depends on a choice of $f$-divergence. When similarly aligned with TV distance, DrilDICE consistently outperforms all baselines.
>
> - **Comparison on Lower-Quality Datasets**:
>     - We conduct additional experiments on segmented trajectory scenario (Scenario 3) with a medium-quality dataset, instead of the expert dataset.
>
> Please refer to our detailed rebuttal for more comprehensive information. If you have any further questions or concerns, please do not hesitate to share your comments.
>
> We sincerely thank you again for your reviews.

---

### Official Review · Reviewer_oZLc · 2024-07-09

**Soundness:** 3
**Presentation:** 3
**Contribution:** 2
**Rating:** 5
**Confidence:** 3

**Summary:**

This paper studies imitation learning when the offline dataset does not come from the stationary expert distribution. To address this problem, the authors introduce the objective of distributionally robust optimization into behavioral cloning. To avoid overly pessimistic solutions, the authors further incorporate the Bellman flow constraint and empirical distribution divergence into the min-max optimization objective. Based on the DICE technology, the solution can be derived. The experimental results have shown the proposal's effectiveness on corrupted datasets.

**Strengths:**

The proposed solution is technically reasonable and theoretically sound.

**Weaknesses:**

1. I agree this paper is well-contained. This work successfully introduces the min-max formulation of DRO to the imitation learning community. However, the new setting proposed in the paper is not well-supported. The shift between the offline dataset and the stationary expert distribution is a reasonable setting. However, all experiments are validated based on simulated shifts, which somewhat makes me feel less excited.
2. The experiments are conducted on simulated covariate shifts. More evaluation in real-world cases will further improve this paper.
3. Some notations have not been well defined, the $\Delta S$ in Equation 2, and the $\mathcal{W}$ in Equation 5.

**Questions:**

In the navigation task like the current experiments in the Four Rooms environment, limited expert trajectories from some start points, especially, have only partial coverage on the state space, is there also a covariate shift with the stationary distribution? I personally think this might be a more real and direct case in real-world applications. Is the method proposed still effective in this case?

**Limitations:**

The authors have provided a discussion about the limitations and broader societal impacts.

---

> ### Author Rebuttal · Authors · 2024-08-07
>
> Thank you for your valuable feedback.
>
> **1. Experimental Design and Real-World Applicability Concerns**
>
> Our experimental setup, though based on simulated data, is intended to rigorously assess the adaptability and robustness of our approach under realistic conditions. This strategy also ensures reproducibility and is consistent with established methodologies in the field, as demonstrated by similar studies like Yan et al., which also employs segments of the D4RL dataset to test hypotheses in learning from observations (LfO) problem setting [A].
>
> Beyond the simulated shifts, we also have conducted tests on complete trajectories with limited data sizes—a scenario frequently encountered in real-world applications. The results, illustrated in **Figure C**, demonstrate that DrilDICE adeptly handles sampling errors in small datasets and surpasses existing approaches in performance.
>
> [A] Yan et al., “A Simple Solution for Offline Imitation from Observations and Examples with Possibly Incomplete Trajectories,” NeurIPS 2023.
>
>
> **2. Clarification on Unclear Notations**
>
> We clarify that $\Delta S$ represents a set of arbitrary state distributions, and $\mathcal{W}$ denotes a function space of $w(s,a)$ with $w:\mathcal{S}\times\mathcal{A} \to \mathbb{R}$. These updates will be reflected in the final version of the paper.
>
> **3. Applicability in Partial Coverage Scenarios**
>
> DrilDICE, like other DICE approaches, relies on the assumption that the support of the target state distribution $d$ should encompass the support of the expert's stationary distribution $d_E$.
> This assumption presents a challenge in ensuring the robustness of DICE approaches when the support of $d$ does not fully cover that of $d_E$.
>
> Despite this, DrilDICE has shown strong performance in scenarios with limited data coverage.
> In our experiments, particularly in Scenario 3 and the complete trajectories experiment (see **Figure B, C** in the PDF respectively), we varied dataset sizes to adjust data coverage. The result suggests that DRO approaches possibly handle sampling errors in small datasets.
>
> While these results are encouraging, the comprehensive applicability of our method when $d$ partially covers $d_E$ is yet to be fully established. We believe that further exploration of this critical aspect represents a promising research topic.
>
>
> **4. Expanded Evaluation Results on Main Scenarios**
>
> We have updated our main experiments to include a baseline DR-BC [18] and a variation of $f$-divergence for our approach accordingly: Inspired by the choice of $f$ in DR-BC, we introduced **the soft-TV distance**, where the $f$ function is a log-cosh function and its derivative is the tanh function. [B] This provides a relaxed and invertible version of TV distance, and enables us to obtain a closed form solution of $w$ by using Proposition 1.
>
> Based on this, we have expanded our analysis by incorporating three additional methods: **DR-BC** for all scenarios, **OptiDICE-BC (with soft-TV)**, and **DrilDICE (with soft-TV)** for Mujoco scenarios. The expanded results for Four Rooms environment and Scenario 1,2 can be found in **Table A, B** and **C** in the comment, and the result for Scenario 3 can be found in **Figure B** in the PDF.
>
> The results consistently show that DrilDICE with the soft-TV outperforms baselines including DR-BC across most scenarios. Notably, DrilDICE with soft-TV, utilizing $f$-divergence similar to the one used in DR-BC, consistently outperforms DR-BC in most evaluation scenarios. We attribute this performance improvement to the remaining key difference: an inclusion of a Bellman flow constraint, which is a key contribution of our work.
>
> The results also demonstrate that the choice of the soft-TV significantly enhances the performance of DrilDICE compared to using soft-$\chi^2$. To explain this performance gain, we hypothesize that the soft-TV distance provides more discriminative weighting $w$ of samples based on their long-term policy errors. As shown in **Figure A** in the PDF, conventional $f$-divergences (e.g. KL, soft-$\chi^2$, …) make $w$ less pronounced to the long-term policy error $e$, yet the soft-TV distance responds sensitively to changes in $e$, resulting in a more pronounced $w$. This enables BC loss to more selectively focus on critical samples with significant magnitude of long-term policy errors, thereby effectively enhancing performance, akin to the benefits observed in Sparse Q-Learning [C].
>
> [B] Saleh et al., "Statistical Properties of the log-cosh Loss Function used in Machine Learning." arXiv preprint (2022).
>
> [C] Xu et al., “Offline RL with No OOD Actions: In-sample Learning via Implicit Value Regularization”, ICLR 2023.
>
>
> Thank you again for your insightful comments. If you have any further concerns or questions about this, please feel free to reply and we will address them promptly.

---

> > ### Comment · Reviewer_oZLc · 2024-08-13
> >
> > Thanks for your response. I agree that segmented trajectories have simulated a type of real-world covariate shift.

---

> > > ### Author Response · Authors · 2024-08-13
> > >
> > > We're glad to hear that some of your concerns have been addressed.
> > >
> > > We truly believe that your insightful feedback has notably improved the quality of our manuscript.
> > >
> > > If you have any remaining concerns, please let us know so that we can further enhance the quality of our research.

---

> > > ### Author Response · Authors · 2024-08-14
> > > **Dear Reviewer oZLc**
> > >
> > > We are awaiting your feedback on any remaining concerns, and we would be very happy to resolve these issues.
> > >
> > > We are eager to address remaining concerns to enhance our manuscript, so please do not hesitate to let us know.
> > >
> > > Thank you once again for your invaluable reviews.

---

> ### Author Response · Authors · 2024-08-07
> **Additional Experimental Results**
>
> ## **Four Rooms Environment**
>
> | Scenario | BC | OptiDICE-BC | DR-BC | DrilDICE (Ours) |
> | --- | --- | --- | --- | --- |
> | Room 1 | 90.84 ± 0.69 | 94.30 ± 0.41 | 91.38 ± 0.73 | **95.04 ± 0.48** |
> | Room 2 | 89.16 ± 1.07 | 94.06 ± 0.65 | 89.28 ± 1.08 | **94.44 ± 0.62** |
> | Room 3 | 88.50 ± 1.27 | 94.20 ± 0.98 | 88.70 ± 1.26 | **95.04 ± 0.86** |
> | Room 4 | 90.94 ± 0.75 | 94.26 ± 0.54 | 90.94 ± 0.75 | **94.92 ± 0.37** |
> | Action UP | 84.96 ± 1.33 | 92.06 ± 0.69 | 84.96 ± 1.33 | **93.22 ± 0.61** |
> | Action DOWN | 89.96 ± 0.96 | 93.62 ± 0.62 | 89.96 ± 0.96 | **94.60 ± 0.39** |
> | Action LEFT | 90.18 ± 1.11 | 91.86 ± 1.03 | 90.18 ± 1.11 | **92.62 ± 0.95** |
> | Action RIGHT | 93.04 ± 0.63 | 94.46 ± 0.44 | 93.44 ± 0.60 | **94.52 ± 0.44** |
>
> **Table A.** Expanded performance comparison of normalized scores on Four Rooms environment. (corresponds to Table 2)
>
> ## **Scenario 1: Rebalanced Dataset**
>
> | Scenario | Task | $p(D_1)$ | BC | OptiDICE-BC (Soft-$\chi^2$) | DrilDICE (Soft-$\chi^2$) | DR-BC | OptiDICE-BC (Soft-TV) | DrilDICE (Soft-TV) |
> | --- | --- | --- | --- | --- | --- | --- | --- | --- |
> | Rebalanced by state | hopper | 0.1 | 24.65 ± 4.15 | 35.26 ± 3.19 | **58.92 ± 4.30** | 27.02 ± 4.31 | 12.72 ± 1.27 | 52.22 ± 5.57 |
> |  |  | 0.5 | 35.38 ± 4.08 | 24.41 ± 5.12 | 60.91 ± 7.61 | 36.71 ± 3.56 | 8.61 ± 1.97 | **67.12 ± 8.18** |
> |  |  | 0.9 | 11.23 ± 2.49 | 17.26 ± 2.10 | 28.29 ± 3.03 | 27.37 ± 4.89 | 10.44 ± 1.76 | **36.39 ± 6.10** |
> |  | walker2d | 0.1 | 18.85 ± 4.05 | 5.91 ± 0.59 | 31.92 ± 5.94 | 14.69 ± 3.26 | 4.91 ± 1.08 | **51.55 ± 8.16** |
> |  |  | 0.5 | 22.91 ± 3.13 | 11.12 ± 1.36 | 30.53 ± 3.47 | 45.08 ± 9.98 | 8.09 ± 0.42 | **73.74 ± 5.37** |
> |  |  | 0.9 | 30.42 ± 7.05 | 17.51 ± 2.61 | 43.31 ± 4.54 | 46.04 ± 8.35 | 7.69 ± 0.45 | **77.60 ± 5.45** |
> |  | halfcheetah | 0.1 | 49.31 ± 5.16 | 33.42 ± 4.70 | 44.79 ± 5.16 | 32.85 ± 3.82 | 7.05 ± 1.47 | **52.45 ± 3.62** |
> |  |  | 0.5 | 37.98 ± 3.07 | 33.36 ± 3.04 | 41.13 ± 3.53 | 26.16 ± 4.91 | 6.14 ± 1.21 | **55.04 ± 3.27** |
> |  |  | 0.9 | 15.54 ± 3.06 | 2.21 ± 1.16 | 7.28 ± 1.59 | 8.96 ± 3.26 | 1.02 ± 1.12 | **22.28 ± 2.88** |
> | Rebalanced by action | hopper | 0.1 | 29.71 ± 4.00 | 28.37 ± 1.11 | 42.29 ± 6.39 | 25.92 ± 2.45 | 11.71 ± 2.12 | **56.60 ± 11.90** |
> |  |  | 0.5 | 26.35 ± 4.88 | 30.03 ± 4.23 | 53.37 ± 9.50 | 35.13 ± 5.41 | 11.79 ± 1.14 | **73.80 ± 3.63** |
> |  |  | 0.9 | 30.50 ± 3.60 | 38.92 ± 4.78 | **63.14 ± 7.12** | 36.56 ± 2.29 | 19.42 ± 2.82 | 48.99 ± 12.27 |
> |  | walker2d | 0.1 | 23.61 ± 5.10 | 12.06 ± 1.20 | 40.72 ± 1.13 | 31.18 ± 4.23 | 7.27 ± 0.46 | **70.60 ± 3.21** |
> |  |  | 0.5 | 32.29 ± 6.74 | 16.40 ± 1.70 | 47.93 ± 12.36 | 30.52 ± 3.89 | 6.37 ± 0.97 | **72.09 ± 8.70** |
> |  |  | 0.9 | 16.87 ± 2.80 | 15.64 ± 3.57 | 43.68 ± 11.50 | 37.55 ± 8.97 | 4.60 ± 0.97 | **69.51 ± 8.54** |
> |  | halfcheetah | 0.1 | 41.91 ± 4.80 | 26.62 ± 2.54 | 32.76 ± 1.87 | 27.50 ± 1.01 | 8.41 ± 3.38 | **56.42 ± 4.57** |
> |  |  | 0.5 | 45.80 ± 4.45 | 45.52 ± 3.24 | 48.08 ± 5.50 | 33.39 ± 6.52 | 4.64 ± 0.84 | **60.81 ± 1.56** |
> |  |  | 0.9 | 25.91 ± 3.35 | 4.28 ± 1.52 | 9.57 ± 2.34 | 12.08 ± 2.01 | 0.59 ± 0.68 | **29.19 ± 4.58** |
>
> **Table B.** Expanded performance comparison on Scenario 1 (rebalanced dataset). (corresponds to Table 3)
>
>
> ## **Scenario 2: Time-dependently Subsampled Dataset**
>
> | Task | (a, b) | BC | OptiDICE-BC (Soft-$\chi^2$) | DrilDICE (Soft-$\chi^2$) | DR-BC | OptiDICE-BC (Soft-TV) | DrilDICE  (Soft-TV) |
> | --- | ------- | --- | --- | --- | --- | --- | --- |
> | hopper | (1, 1) | 28.89 ± 3.77 | 50.33 ± 6.60 | **54.83 ± 7.66** | 21.10 ± 2.26 | 22.77 ± 3.94 | 45.44 ± 5.11 |
> |  | (1, 5) | 31.03 ± 0.90 | 39.54 ± 4.02 | 37.18 ± 7.92 | 25.00 ± 1.66 | 19.25 ± 1.21 | **45.60 ± 4.63** |
> |  | (5, 1) | 26.75 ± 7.12 | **48.40 ± 12.98** | 39.91 ± 9.20 | 17.51 ± 3.38 | 25.68 ± 6.01 | 34.71 ± 9.00 |
> |  | (5, 5) | 27.65 ± 6.71 | 32.46 ± 10.79 | **40.24 ± 7.41** | 23.20 ± 6.32 | 14.12 ± 3.59 | 25.61 ± 6.03 |
> | walker2d | (1, 1) | 28.95 ± 5.34 | 17.42 ± 3.00 | 51.85 ± 5.30 | 45.66 ± 9.92 | 6.13 ± 1.03 | **81.21 ± 5.40**|
> |  | (1, 5) | 61.48 ± 5.19 | 37.25 ± 5.66 | 64.46 ± 7.92 | 57.29 ± 4.79 | 17.55 ± 2.65 | **84.28 ± 4.89** |
> |  | (5, 1) | 8.13 ± 0.72 | 4.43 ± 0.91 | 23.31 ± 3.44 | 17.97 ± 2.58 | 4.37 ± 0.52 | **48.23 ± 8.30** |
> |  | (5, 5) | 6.65 ± 1.20 | 8.50 ± 2.27 | 14.40 ± 3.21 | 12.45 ± 1.84 | 5.54 ± 0.68 | **52.57 ± 6.24** |
> | halfcheetah | (1, 1) | 33.74 ± 2.99 | 33.65 ± 5.49 | 33.43 ± 3.73 | 17.09 ± 2.43 | 9.93 ± 3.56 | **44.17 ± 5.62** |
> |  | (1, 5) | 72.72 ± 2.60 | 52.94 ± 3.98 | 69.63 ± 3.86 | 61.81 ± 2.50 | 24.42 ± 3.27 | **77.12 ± 2.42** |
> |  | (5, 1) | 2.35 ± 0.51 | 2.46 ± 1.05 | 3.97 ± 1.18 | 3.81 ± 1.26 | 1.29 ± 1.13 | **5.68 ± 1.50** |
> |  | (5, 5) | 2.01 ± 0.91 | 1.61 ± 1.30 | 4.61 ± 1.55 | 2.85 ± 1.05 | -1.19 ± 0.37 | **5.50 ± 0.83** |
>
> **Table C.** Expanded performance comparison on Scenario 2 (time-dependently collected dataset). (corresponds to Table 4)

---

> ### Author Response · Authors · 2024-08-12
> **Dear Reviewer oZLc**
>
> We gently remind you that less than 48 hours remain in our discussion period. We are committed to thoroughly addressing the reviewer’s remaining concerns. In summary, our responses are as follows:
>
> - **Concerns on Experiments on Simulated Shift**:
>     - Our covariate shift experiment setting ensures robustness and reproducibliity of comparison. Similar simulated shifts have been utilized in exisiting literature.
>     - Additionally, we have included natural complete trajectory scenarios where DrilDICE shows significant data-efficiency.
>
> - **Concerns on Partial Coverage**:
>    - Although DrilDICE is not specifically designed to address partial coverage issues, it has shown empirical robustness in such scenarios.
>
> - **Expanded Main Experiments**:
>     - We introduced DR-BC as a new baseline across all experiments; DrilDICE consistently outperforms DR-BC when selecting a similar $f$-divergence.
>
> Please see our rebuttal for more information. If you have any further questions or concerns, do not hesitate to share your comments.
>
> Again, thank you for your thoughtful reviews.

---

### Official Review · Reviewer_zQPx · 2024-07-09

**Soundness:** 2
**Presentation:** 3
**Contribution:** 3
**Rating:** 6
**Confidence:** 4

**Summary:**

This paper devices distribution correction ratio estimation (DICE)-based optimization to mitigate the covariate shift issue in the Behavior cloning algorithm. They test their heuristic loss on Mujoco benchmarks.

**Strengths:**

The design of drildice in Section 3.2 and its evaluation on Mujoco benchmark are the strengths of this work.

**Weaknesses:**

The results in this work are preliminary and need to be evaluated carefully. I move all my concerns to the Questions section below instead of treating these as weaknesses. I will also rely on the author-reviewer and reviewer-reviewer discussion periods for updating my decisions.

**Questions:**

- Section 3.2 requires further details and refined writing. For example,
  - Slater's condition/assumption must be explicitly mentioned since the drildice algorithm relies on it.
  - Example of $f$ and $f'$ should be mentioned in this section to make sense of the experiments section (they are only mentioned in Appendix as KL and $\chi^2$.)
  - More details are needed to support this statement: "the problem can be solved by alternatively optimizing $\nu$ and $\pi$." Such as citing previous DICE works and so on. E.g reference https://proceedings.neurips.cc/paper_files/paper/2019/file/cf9a242b70f45317ffd281241fa66502-Paper.pdf
  - These details will also help in providing some theoretical justifications from previous DICE-related works, which this paper currently lacks.

- Comparison with previous works must be further expanded. Especially with [18].
  - Both this paper Section 3.1 and [18] share similar motivation to come up with loss functions to mitigate covariate shift created by $d_D\neq d_E$.
  - At line 92, it is mentioned that $d_D\neq d_E$ but the underlying policy (which is $\pi_E$) is the same for the two state-action visitation distributions. Now, considering the definition of state-action visitation distribution at line 54, it is easy to see that $d_D$ and $d_E$ in fact only differ in the shifts corresponding to transition distribution T of the underlying MDP. Thus, this paper is considering transition distribution shifts as opposed to the claim at line 89!
  - Based on the above points, [18] also considers distribution matching in their inner problem w.r.t transition distribution T shifts.
  - Despite this similarity between [18] and this work, both take different approaches to design the loss. This work takes the Bellman flow constraints route whereas [18] considers direct DRO on the transition distribution T shifts.
  - [18] provides both theoretical guarantees and empirical evaluations and this paper provides only empirical evaluations.

- Regarding experiments
  - Can you please explain how $D_i$ at line 194 differs in size of the dataset compared to $D_E$? With $p(u)=0.4$, is it the case that 40% of states are sampled from $D_E$ and the rest are random states?
  - The analogy of sampling in line 187 "if a data collection device (e.g. cameras or sensors) operates at different recording frequencies in each room" can also be considered as shifts in transition distribution T, thus sharing the similarities as discussed before. Similar observation can be made for the Scenarios 1 and 3. Connection to transition shifts is more apparent in Scenario 2. So I think it is worthwhile adding more benchmark algorithms [2, 18].
  - Can you also include another scenario, training without covariate shift for instance? BC and other non-robust versions should outperform in such ablation-type tests.

My score reflects this review provided here.

---

> ### Author Rebuttal · Authors · 2024-08-07
>
> First and foremost, we sincerely appreciate your insightful feedback.
>
> **1. Justification on Alternative Optimization**
>
> To clarify, our initial statement suggested that alternating optimization provides a practical approach to approximating the solution of the problem as outlined in Equation (7). In response, we will clearly revise this statement to *“... the problem can be practically addressed by alternately optimizing $\nu$ and $\pi$, following recent developments in DICE approaches. [citations]”* with related literature including suggested DualDICE and OptiDICE. Additionally, we will provide a more detailed exposition on the derivation (e.g. details on Slater’s condition, assumptions) in the forthcoming revision.
>
> **2. Clarification on $d_D$ and Our Problem Setting**
>
> It is important to note that we did not assume that $d_D$ is a stationary distribution of any policy; rather, it is an arbitrary state distribution. Hence, even if the transition dynamics $T$ and the policy $\pi_E$ remain unchanged, $d_D$ can deviate from $d_E$ since we do not assume that $d_D$ should be induced from MDPs. We clarify this point in the revision.
>
> Additionally, we also clarify that our problem setting concerns shifts in stationary distribution (say $d$-shifts), not transition shifts ($T$-shifts). Specifically, we assumed that the expert dataset is a collection of transitions $(s,a,s’)$, where $s$ sampled from $d_D$, an expert action $a$ decided by the deterministic expert $\pi_E$, and $s’$ sampled from $T(s’|s,a)$ (which will not be shifted in the testing phase). Even when $T$ and $\pi_E$ remain unchanged, practical constraints such as high costs or limited recording frequencies often result in dataset with sparsely collected transitions. This sparsity leads to incomplete trajectories, causing a deviation of $d_D(s)$ from $d_E(s)$. We do not focus on delayed transitions such as ($s_t, a_t, s_{t+k}$) with $k \ge 2$, which may arise from not collecting immediate subsequent states.
>
> Although we maintain that our setting does not directly involve $T$-shifts, we compare our approach with DR-BC, as their objective also pertains to $d$-shifts.
>
> **3. Comparison with DR-BC [18]**
>
> In response to the reviewer’s suggestion, we have revisited DR-BC [18] and investigated the method more thoroughly. While DR-BC primarily aims to address $T$-shift scenarios, its objective considers the uncertainty set of arbitrary state distributions $d(s)$, which is potentially applicable to our scenario. Hence, we evaluate DR-BC as a baseline for all of our scenarios. To compare, the key differences between DR-BC and our approach can be summarized as:
>
> - **Choice of Uncertainty Set**:  To define its uncertainty set, DR-BC employs TV distance while DrilDICE uses $f$-divergence, which is more general.
> - **Adoption of Bellman Flow Constraints**: For an uncertainty set, DR-BC considers *arbitrary state distributions*, including non-stationary ones, while DrilDICE considers *state stationary distributions* by enforcing Bellman flow constraints.
>
> **4. Technical Adjustment for DrilDICE and Summary of $f$-divergence**
>
> We initially attempted to align the $f$-divergence to the TV distance used in DR-BC for a fair comparison.
> Our formulation requires the inverse of the derivative of $f$ for a closed form solution of the weight $w$.
> However, since the derivative $f$ of TV distance is a step function and is not invertible, we cannot directly use TV distance.
> We address this issue by adopting the log-cosh function [A] for $f$, which has a tanh function as $f’$; a relaxed version of the step function while ensuring invertibility.
> We refer to this as **the soft-TV distance** and utilized this in Mujoco-based experiments. (See **Figure A** for visualization)
>
> In summary, the following table compares the choice of $f$-divergence for DrilDICE, highlighting our method's adaptability in obtaining practical solutions through refined mathematical approaches.
> As the reviewer commented, we will include this table in the main manuscript.
>
> | Divergence | $f(x)$ | $(f')^{-1}(y)$ | $w^*_{\pi,\nu}$ |
> | --- | --- | --- | --- |
> | KL Divergence | $x\log x$ | $\exp(y-1)$ | $\exp\left(\frac{e_{\pi,\nu}(s,a, s')}{\alpha}-1\right)$ |
> | $\chi^2$-Divergence | $\frac{1}{2}(x-1)^2$ | $(y+1)$ | $\mathrm{ReLU}(\frac{e_{\pi,\nu}(s,a, s')}{\alpha} + 1)$ |
> | Soft-$\chi^2$ Divergence | $f_{\text{soft}-\chi^2} (x) $ | $(f^{'})^{-1}_{\text{soft}-\chi^2} (y) $ | $\mathrm{ELU}\left(\frac{e_{\pi,\nu}(s,a, s')}{\alpha}\right) + 1$ |
> | TV Distance | $\frac{1}{2}\|x-1\|$ | - | - |
> | Soft-TV Distance | $\frac{1}{2} \log (\cosh( x-1))$ | $\tanh^{-1}(2y)+1$ | $\mathrm{ReLU}(\tanh^{-1}(\frac{2e_{\pi,\nu}(s,a, s')}{\alpha})+1)$ |
>
> where $f_{\text{soft}-\chi^2} (x) := x\log x -x +1$ if $0<x<1$ and $(x-1)^2$ if $x \ge 1$,
> $(f^{'})^{-1}_{\text{soft}-\chi^2} (y) := \exp(y)$ if $y<0$ and $y+1$ if $y \ge 0$,
> $\mathrm{ReLU}(x):=\max(0,x)$, $\mathrm{ELU}(x):= \exp(x)-1$ if $x<0$ and $x$ for $x\ge0$.
>
> [A] Saleh et al., "Statistical Properties of the log-cosh Loss Function used in Machine Learning." arXiv preprint (2022).

---

> ### Author Response · Authors · 2024-08-07
> **Rebuttal (continued)**
>
> **5. Expanded Results on Main Experiments**
>
> We added three additional methods: **DR-BC** for all scenarios, **OptiDICE-BC (with soft-TV)**, **DrilDICE (with soft-TV)** for Mujoco scenarios. The expanded results for Four Rooms environment and Scenario 1,2 are presented respectively in **Table A, B**, and **C** respectively. The results for Scenario 3 are detailed in **Figure B** of the PDF.
>
> The results consistently show that DrilDICE with the soft-TV outperforms baselines including DR-BC across most scenarios. Notably, DrilDICE with soft-TV, utilizing $f$-divergence similar to the one used in DR-BC, consistently outperforms DR-BC in most evaluation scenarios. We attribute this performance improvement to the remaining key difference: an inclusion of a Bellman flow constraint, which is a key contribution of our work.
>
> The results also demonstrate that the choice of the soft-TV significantly enhances the performance of DrilDICE compared to using soft-$\chi^2$. To explain this performance gain, we hypothesize that the soft-TV distance provides more discriminative weighting $w$ of samples based on their long-term policy errors. As shown in **Figure A** in the PDF, conventional $f$-divergences (e.g. KL, soft-$\chi^2$, …) make $w$ less pronounced to the long-term policy error $e$, yet the soft-TV distance responds sensitively to changes in $e$, resulting in a more pronounced $w$. This enables BC loss to selectively focus on critical samples with significant magnitude of long-term policy errors, thereby effectively enhancing performance, akin to the benefits observed in Sparse Q-Learning [B].
>
> [B] Xu et al., “Offline RL with No OOD Actions: In-sample Learning via Implicit Value Regularization”, ICLR 2023.
>
> **6. Clarification on Datasets Used in Four Rooms Experiments**
>
> In our Four Rooms experiment, we designed the covariate shift scenarios caused by unobserved factors affecting data curation, specifically considering such factors as (1) room visitation or (2) action.
> In the Room 1 manipulation scenario, we initially split the expert dataset into two subsets $D_A,D_B$ based on whether each transition’s state was associated with Room 1 or not. We then subsampled transitions from subsets $D_A, D_B$ using the predetermined proportion $p(u), 1-p(u)$, respectively and combined them to construct the shifted dataset $D_i$.
>
> Specifically, we utilized 100 original episodes, comprising a total of 3994 transitions with a maximum episode length of 50, for $D_E$. From these, after setting $p(u) = 0.4$, we subsampled 1000 transitions to construct $D_i$, which includes 40% (400 transitions) from $D_A$, and 60% (600 transitions) from $D_B$ (e.g., transitions in Rooms 2, 3, 4). To avoid potential issues with support coverage—beyond the focus of our study—we ensured that all states  ($|S| = 11 \times 11 = 121$) in $D_i$ appear at least once, as detailed in the Appendix.
>
> **7. Experiments on Standard Scenarios**
>
> We acknowledge that our initial experiments did not include comparisons on standard scenarios.
> To address this, we have conducted additional experiments using complete expert trajectories using
> D4RL `expert-v2` dataset varying the number of trajectories in \{1, 5, 10, 50\}. The results are detailed in **Figure C** in the PDF. We also observed that DrilDICE can deal with sampling errors of small datasets, showing a superior performance compared to other methods.
>
> Once again, we extend our sincere thanks for your valuable feedback. We believe that addressing your concerns has substantially enhanced the quality of our research. We plan to incorporate all our discussions into the final version. If you have any remaining concerns or questions, please do not hesitate to comment and we will respond as soon as possible.

---

> ### Author Response · Authors · 2024-08-07
> **Additional Experimental Results**
>
> ## **Four Rooms Environment**
>
> | Scenario | BC | OptiDICE-BC | DR-BC | DrilDICE (Ours) |
> | --- | --- | --- | --- | --- |
> | Room 1 | 90.84 ± 0.69 | 94.30 ± 0.41 | 91.38 ± 0.73 | **95.04 ± 0.48** |
> | Room 2 | 89.16 ± 1.07 | 94.06 ± 0.65 | 89.28 ± 1.08 | **94.44 ± 0.62** |
> | Room 3 | 88.50 ± 1.27 | 94.20 ± 0.98 | 88.70 ± 1.26 | **95.04 ± 0.86** |
> | Room 4 | 90.94 ± 0.75 | 94.26 ± 0.54 | 90.94 ± 0.75 | **94.92 ± 0.37** |
> | Action UP | 84.96 ± 1.33 | 92.06 ± 0.69 | 84.96 ± 1.33 | **93.22 ± 0.61** |
> | Action DOWN | 89.96 ± 0.96 | 93.62 ± 0.62 | 89.96 ± 0.96 | **94.60 ± 0.39** |
> | Action LEFT | 90.18 ± 1.11 | 91.86 ± 1.03 | 90.18 ± 1.11 | **92.62 ± 0.95** |
> | Action RIGHT | 93.04 ± 0.63 | 94.46 ± 0.44 | 93.44 ± 0.60 | **94.52 ± 0.44** |
>
> **Table A.** Expanded performance comparison of normalized scores on Four Rooms environment. (corresponds to Table 2)
>
> ## **Scenario 1: Rebalanced Dataset**
>
> | Scenario | Task | $p(D_1)$ | BC | OptiDICE-BC (Soft-$\chi^2$) | DrilDICE (Soft-$\chi^2$) | DR-BC | OptiDICE-BC (Soft-TV) | DrilDICE (Soft-TV) |
> | --- | --- | --- | --- | --- | --- | --- | --- | --- |
> | Rebalanced by state | hopper | 0.1 | 24.65 ± 4.15 | 35.26 ± 3.19 | **58.92 ± 4.30** | 27.02 ± 4.31 | 12.72 ± 1.27 | 52.22 ± 5.57 |
> |  |  | 0.5 | 35.38 ± 4.08 | 24.41 ± 5.12 | 60.91 ± 7.61 | 36.71 ± 3.56 | 8.61 ± 1.97 | **67.12 ± 8.18** |
> |  |  | 0.9 | 11.23 ± 2.49 | 17.26 ± 2.10 | 28.29 ± 3.03 | 27.37 ± 4.89 | 10.44 ± 1.76 | **36.39 ± 6.10** |
> |  | walker2d | 0.1 | 18.85 ± 4.05 | 5.91 ± 0.59 | 31.92 ± 5.94 | 14.69 ± 3.26 | 4.91 ± 1.08 | **51.55 ± 8.16** |
> |  |  | 0.5 | 22.91 ± 3.13 | 11.12 ± 1.36 | 30.53 ± 3.47 | 45.08 ± 9.98 | 8.09 ± 0.42 | **73.74 ± 5.37** |
> |  |  | 0.9 | 30.42 ± 7.05 | 17.51 ± 2.61 | 43.31 ± 4.54 | 46.04 ± 8.35 | 7.69 ± 0.45 | **77.60 ± 5.45** |
> |  | halfcheetah | 0.1 | 49.31 ± 5.16 | 33.42 ± 4.70 | 44.79 ± 5.16 | 32.85 ± 3.82 | 7.05 ± 1.47 | **52.45 ± 3.62** |
> |  |  | 0.5 | 37.98 ± 3.07 | 33.36 ± 3.04 | 41.13 ± 3.53 | 26.16 ± 4.91 | 6.14 ± 1.21 | **55.04 ± 3.27** |
> |  |  | 0.9 | 15.54 ± 3.06 | 2.21 ± 1.16 | 7.28 ± 1.59 | 8.96 ± 3.26 | 1.02 ± 1.12 | **22.28 ± 2.88** |
> | Rebalanced by action | hopper | 0.1 | 29.71 ± 4.00 | 28.37 ± 1.11 | 42.29 ± 6.39 | 25.92 ± 2.45 | 11.71 ± 2.12 | **56.60 ± 11.90** |
> |  |  | 0.5 | 26.35 ± 4.88 | 30.03 ± 4.23 | 53.37 ± 9.50 | 35.13 ± 5.41 | 11.79 ± 1.14 | **73.80 ± 3.63** |
> |  |  | 0.9 | 30.50 ± 3.60 | 38.92 ± 4.78 | **63.14 ± 7.12** | 36.56 ± 2.29 | 19.42 ± 2.82 | 48.99 ± 12.27 |
> |  | walker2d | 0.1 | 23.61 ± 5.10 | 12.06 ± 1.20 | 40.72 ± 1.13 | 31.18 ± 4.23 | 7.27 ± 0.46 | **70.60 ± 3.21** |
> |  |  | 0.5 | 32.29 ± 6.74 | 16.40 ± 1.70 | 47.93 ± 12.36 | 30.52 ± 3.89 | 6.37 ± 0.97 | **72.09 ± 8.70** |
> |  |  | 0.9 | 16.87 ± 2.80 | 15.64 ± 3.57 | 43.68 ± 11.50 | 37.55 ± 8.97 | 4.60 ± 0.97 | **69.51 ± 8.54** |
> |  | halfcheetah | 0.1 | 41.91 ± 4.80 | 26.62 ± 2.54 | 32.76 ± 1.87 | 27.50 ± 1.01 | 8.41 ± 3.38 | **56.42 ± 4.57** |
> |  |  | 0.5 | 45.80 ± 4.45 | 45.52 ± 3.24 | 48.08 ± 5.50 | 33.39 ± 6.52 | 4.64 ± 0.84 | **60.81 ± 1.56** |
> |  |  | 0.9 | 25.91 ± 3.35 | 4.28 ± 1.52 | 9.57 ± 2.34 | 12.08 ± 2.01 | 0.59 ± 0.68 | **29.19 ± 4.58** |
>
> **Table B.** Expanded performance comparison on Scenario 1 (rebalanced dataset). (corresponds to Table 3)
>
>
> ## **Scenario 2: Time-dependently Subsampled Dataset**
>
> | Task | (a, b) | BC | OptiDICE-BC (Soft-$\chi^2$) | DrilDICE (Soft-$\chi^2$) | DR-BC | OptiDICE-BC (Soft-TV) | DrilDICE  (Soft-TV) |
> | --- | ------- | --- | --- | --- | --- | --- | --- |
> | hopper | (1, 1) | 28.89 ± 3.77 | 50.33 ± 6.60 | **54.83 ± 7.66** | 21.10 ± 2.26 | 22.77 ± 3.94 | 45.44 ± 5.11 |
> |  | (1, 5) | 31.03 ± 0.90 | 39.54 ± 4.02 | 37.18 ± 7.92 | 25.00 ± 1.66 | 19.25 ± 1.21 | **45.60 ± 4.63** |
> |  | (5, 1) | 26.75 ± 7.12 | **48.40 ± 12.98** | 39.91 ± 9.20 | 17.51 ± 3.38 | 25.68 ± 6.01 | 34.71 ± 9.00 |
> |  | (5, 5) | 27.65 ± 6.71 | 32.46 ± 10.79 | **40.24 ± 7.41** | 23.20 ± 6.32 | 14.12 ± 3.59 | 25.61 ± 6.03 |
> | walker2d | (1, 1) | 28.95 ± 5.34 | 17.42 ± 3.00 | 51.85 ± 5.30 | 45.66 ± 9.92 | 6.13 ± 1.03 | **81.21 ± 5.40**|
> |  | (1, 5) | 61.48 ± 5.19 | 37.25 ± 5.66 | 64.46 ± 7.92 | 57.29 ± 4.79 | 17.55 ± 2.65 | **84.28 ± 4.89** |
> |  | (5, 1) | 8.13 ± 0.72 | 4.43 ± 0.91 | 23.31 ± 3.44 | 17.97 ± 2.58 | 4.37 ± 0.52 | **48.23 ± 8.30** |
> |  | (5, 5) | 6.65 ± 1.20 | 8.50 ± 2.27 | 14.40 ± 3.21 | 12.45 ± 1.84 | 5.54 ± 0.68 | **52.57 ± 6.24** |
> | halfcheetah | (1, 1) | 33.74 ± 2.99 | 33.65 ± 5.49 | 33.43 ± 3.73 | 17.09 ± 2.43 | 9.93 ± 3.56 | **44.17 ± 5.62** |
> |  | (1, 5) | 72.72 ± 2.60 | 52.94 ± 3.98 | 69.63 ± 3.86 | 61.81 ± 2.50 | 24.42 ± 3.27 | **77.12 ± 2.42** |
> |  | (5, 1) | 2.35 ± 0.51 | 2.46 ± 1.05 | 3.97 ± 1.18 | 3.81 ± 1.26 | 1.29 ± 1.13 | **5.68 ± 1.50** |
> |  | (5, 5) | 2.01 ± 0.91 | 1.61 ± 1.30 | 4.61 ± 1.55 | 2.85 ± 1.05 | -1.19 ± 0.37 | **5.50 ± 0.83** |
>
> **Table C.** Expanded performance comparison on Scenario 2 (time-dependently collected dataset). (corresponds to Table 4)

---

> ### Author Response · Authors · 2024-08-12
> **Dear Reviewer zQPx**
>
> We kindly remind you that less than 48 hours remain in our discussion period. We are committed to addressing any remaining concerns.
>
> In summary, we have addressed your key concerns as follows:
>
> - **Clarification of Problem Setting**: (1) we do not assume that the data distribution $d_D$ is not stationary, (2) we consider shifts of $d$, not $T$.
> - **Comparison with DR-BC [18]**: (1) a choice of $f$-divergence for the uncertainty set, (2) our method includes Bellman flow constraints.
> -  **Additional Experiments**:
>     - DR-BC has been added as a baseline across all experiments. DrilDICE consistently outperforms DR-BC under similar choices of $f$-divergence.
>     - We have incorporated complete trajectory scenarios, demonstrating significant data efficiency in DrilDICE.
>
> For further information, please see our rebuttal. If you have any additional questions or concerns, we encourage you to provide your comments.
>
> Thank you again for your insightful reviews.

---

> > ### Comment · Reviewer_zQPx · 2024-08-12
> >
> > Thank you for reflecting on the reviews. I have updated my score from 4 to 6. Good luck.

---

> > > ### Author Response · Authors · 2024-08-12
> > >
> > > We sincerely thank the reviewer for the thoughtful review and positive assessment of our work!
> > >
> > > We believe that your insights clearly enhance the quality of our manuscript. We will incorporate your valuable feedback and suggestions into the next revision.

---

### Author Rebuttal · Authors · 2024-08-07

# General Response

We are grateful for the insightful and detailed feedback provided by all reviewers. Below, we summarize our response to the main concerns raised. Should any points require further clarification or detailed discussion, we are fully prepared to engage in discussions during the author-reviewer discussion period.

## **1. Comparison with DR-BC [18]**
Thanks to reviewer zQPx’s suggestion, we found that DR-BC’s objective is also related to our problem setting.
In short, DR-BC considers the uncertainty set of arbitrary state distributions by utilizing TV distance. We evaluated DR-BC as a baseline for all scenarios.

## **2. Soft-TV Distance: Technical Adjustment of $f$ for DrilDICE**
To provide a fair comparison between DrilDICE and DR-BC, we introduced **the soft-TV distance** into DrilDICE. This relaxed version of the TV distance has an invertible derivative of $f$, enabling DrilDICE to obtain a closed form solution of $w^*_{\pi,\nu}$ while maintaining similar properties of TV distance. See **Figure A** in the PDF to compare with other $f$-divergence.

## **3. Additional Experimental Results**
We have expanded our main experiments by incorporating three additional methods: DR-BC, OptiDICE-BC (w\ soft-TV) and DrilDICE (w\ soft-TV) for our main scenarios. Due to space constraints, we omitted the results for DemoDICE, AW-BC, worst-25% score, target 0-1 loss in tables. For the expanded main experiments, please refer to the following:
- Four Rooms : **Table A** in the comment to the rebuttal
- Scenario 1 (rebalanced dataset) : **Table B** in the comment to the rebuttal
- Scenario 2 (time-dependently subsampled dataset) : **Table C** in the comment to the rebuttal
- Scenario 3 (segmented trajectory dataset) : **Figure B** in the PDF

Additionally, we conducted experiments in the following additional problem settings:
- Complete trajectory setting to address reviewer zQPx’s concern : **Figure C** in the PDF
- Medium-quality segment setting as commented by reviewer k6J7 : **Figure D** in the PDF

In summary, the results of additional experiments consistently demonstrate that using the soft-TV distance as an $f$-divergence significantly improves the performance of DrilDICE, outperforming baselines including DR-BC.

---

### Decision · Program_Chairs · 2024-09-25

**Decision:**

Accept (poster)

**Comment:**

As there is a consensus on the acceptance, I would recommend it. It would be great if the author could add some more theoretical analysis (finite sample analysis) to see how we really address covariate shifts in a fundamental way.
(Nitpick: I would recommend following the standard rule (not using official comments) in the rebuttal phase from next time)